# NASICON-type air-stable and all-climate cathode for sodium-ion batteries with low cost and high-power density

Mingzhe Chen [1], Weibo Hua[2], Jin Xiao [3,4], David Cortie[1], Weihua Chen[5], Enhui Wang[1,6], Zhe Hu[1], Qinfen Gu [7], Xiaolin Wang[1], Sylvio Indris[2], Shu-Lei Chou [1] & Shi-Xue Dou [1]

The development of low-cost and long-lasting all-climate cathode materials for the sodium ion battery has been one of the key issues for the success of large-scale energy storage. One option is the utilization of earth-abundant elements such as iron. Here, we synthesize a NASICON-type tuneable $Na_4Fe_3(PO_4)_2(P_2O_7)$/C nanocomposite which shows both excellent rate performance and outstanding cycling stability over more than 4400 cycles. Its air stability and all-climate properties are investigated, and its potential as the sodium host in full cells has been studied. A remarkably low volume change of 4.0% is observed. Its high sodium diffusion coefficient has been measured and analysed via first-principles calculations, and its three-dimensional sodium ion diffusion pathways are identified. Our results indicate that this low-cost and environmentally friendly $Na_4Fe_3(PO_4)_2(P_2O_7)$/C nanocomposite could be a competitive candidate material for sodium ion batteries.

[1] Australian Institute for Innovative Materials, Institute for Superconducting and Electronic Materials, University of Wollongong, Innovation Campus, Squires Way, North Wollongong, NSW 2522, Australia. [2] Institute for Applied Materials-Energy Storage Systems (IAM-ESS), Karlsruhe Institute of Technology (KIT), 76344 Eggenstein-Leopoldshafen, Germany. [3] School of Science, Hunan University of Technology, Zhuzhou 412007, P. R. China. [4] State Key Laboratory of the Superlattices and Microstructures Institute of Semiconductors, Chinese Academy of Sciences, Beijing 100083, P. R. China. [5] College of Chemistry and Molecular Engineering, Zhengzhou University, Zhengzhou 450001, P. R. China. [6] College of Chemical Engineering, Sichuan University, Chengdu 610065, P. R. China. [7] Australian Synchrotron, 800 Blackburn Road, Clayton, VIC 3168, Australia. Correspondence and requests for materials should be addressed to W.C. (email: chenweih@zzu.edu.cn) or to S.-L.C. (email: shulei@uow.edu.au)

Cost-efficient large-scale energy storage systems (EESs) have been in high demand in recent years due to the rapid development of renewable energy resources (i.e., solar, wind, geothermal, and tidal energy)[1,2]. Although advanced lithium-ion battery (LIBs) technology has led to commercially viable electric vehicles (EVs) and provided a satisfactory energy density level for large-scale EESs, the issues of lithium costs and the uneven global distribution of lithium resources have hindered their further application in the large-scale energy storage field[3,4]. Sodium-ion batteries (SIBs) have been considered as promising candidates for EESs because of the high abundance, wide availability, and low cost of sodium resources[5,6]. To achieve high-performance SIBs, however, is still a great challenge, because it is necessary to discover or develop more suitable electrodes to meet the requirements of high energy density and high cycling stability in the most cost-efficient way[7]. The use of the redox chemistry of earth-abundant transition metals, however, will greatly reduce manufacturing costs, which will boost the real applications of SIBs in the commercial EES market[8–10].

The polyanionic or mixed-polyanion system is one of the most important candidates among the various types of electrode materials suitable for SIBs, and it has been the subject of extensive investigations for applications in recent years[11]. The three-dimensional (3D) framework of such electrodes can provide strong and lasting structural support for repeated $Na^+$ ion de-/insertions with relatively high operating potentials. Several vanadium-based compounds have been widely studied as an important group of promising electrodes, such as sodium superionic conductor (NASICON)-type $Na_3V_2(PO_4)_3$[12,13], $Na_3(VO_x)_2(PO_4)F_{3-2x}$ ($x = 0$ or 1)[14,15], $Na_7V_4(P_2O_7)_4PO_4$[16,17], etc. These compounds exhibit satisfactory high energy densities that are comparable to those of LIBs, based on multi-electron redox reactions ($V^{3+}/V^{4+}$ and $V^{4+}/V^{5+}$) and high operating voltage, although the use of toxic and expensive V element remains a critical issue in real applications. Other electrodes based on costly 3d transition metal elements such as Ni-based or Co-based electrodes also face this problem. Therefore, the most abundant and non-toxic 3d element, iron, is the first choice as the redox centre in the polyanionic or mixed-polyanion system[18]. The Fe-based polyanionic compounds have shown high structural stability with small volume change during cycling and sufficiently long cycling stability with high energy density, which will definitely reduce the overall electrode cost and promote the real application of SIBs in large-scale EESs in the near future[8,19–21].

Another critical issue for both LIBs and SIBs is their performance and reliability at the temperatures that might be encountered under all-climate conditions. The commercial $LiFePO_4$ material only can deliver ~70 % of its theoretical capacity when the temperature is reduced to −20 °C, which significantly lowers its applicable energy density[22,23]. SIBs face an even more severe low-temperature problem due to their intrinsically more sluggish solid-diffusion process compared with LIBs[24]. Also, at high temperatures (over 40 °C), some cathodes will encounter inevitable capacity loss and degradation of their cycling stability, such as layered oxide materials for both LIBs and SIBs[25,26]. Cathode $Na^+$-host materials operated at room temperature have been extensively investigated, but their all-climate performance has received little attention in recent years[7,27,28]. For real applications of SIBs in the commercial market, all-climate performance is strongly required, because the energy storage systems must work over a wide range of atmospheric temperatures. Therefore, finding an environmentally friendly, low-cost, and all-climate cathode material is very important for the real application of SIBs.

With these considerations in mind, we screened a variety of iron-based polyanionic materials, and the recently reported mixed crystalline framework structure represented by the ortho-pyrophosphates, such as $Na_4Fe_3(PO_4)_2P_2O_7$, came to our attention[29–32]. $Na_4Fe_3(PO_4)_2P_2O_7$ has not been well studied, but it possesses unexpected 3D sodium diffusion pathways in the sturdy crystal framework of a typical NASICON-type structure, which can be considered as the key preliminary factor for long-term cyclability and low-temperature performance[5,19].

Here, in this paper, we report a low-cost NASICON-type cathode material, $Na_4Fe_3(PO_4)_2(P_2O_7)$, with favourable Na storage properties, that demonstrates fast and stable electrochemical properties under all-climate temperatures, combined with tuneable carbon-coated nanoparticles to form a robust composite material without losing its crystallinity. The uniformly carbon-coated nanosized particles provide facile and rapid electron transport along with high ionic diffusion capability. Excellent rate performances were achieved, and impressive cycling stability was obtained at both room temperature and low/high temperature (−20 °C/50 °C). In addition, we found that this material is stable in air, even after exposure for three months, and Fe-based full SIB configurations are demonstrated using $Fe_3O_4$ nanospheres/hard carbon as anode. In-situ synchrotron X-ray diffraction (XRD) and in-situ X-ray absorption near-edge structure (XANES) analyses further revealed the outstanding reversibility of $Na_4Fe_3(PO_4)_2(P_2O_7)$ in detail. We also employed density functional theory (DFT) studies as well as bond valence sum (BVS) calculations to examine the details of each possible sodium diffusion pathway. Our results indicate that this new NASICON-type $Na_4Fe_3(PO_4)_2(P_2O_7)$ material is a promising cathode candidate for the next generation of SIBs in the near future.

## Results

**Materials characterizations.** Both nanosized $Na_4Fe_3(PO_4)_2(P_2O_7)$ plates (denoted as NFPP-E) and microporous $Na_4Fe_3(PO_4)_2(P_2O_7)$ particles (denoted as NFPP-C) materials were successfully synthesized via a facile one-step sol–gel method. The detailed synthesis procedures for both samples can be found in the experimental section. In order to obtain accurate phase information, high-resolution synchrotron powder X-ray diffraction (s-XRD) was employed to determine the phase constitution and atomic site occupations. Figure 1a shows the satisfactory Rietveld refinement of NFPP-E with a good weighted profile R-factor ($R_{wp} = 6.97\%$). The sample was indexed to the orthorhombic $Pn2_1a$ space group, as previously reported by Kang's group[29], with lattice parameters $a = 17.6433(6)$ Å, $b = 6.3616(4)$ Å, $c = 10.2043(4)$ Å, and $V = 1145.342(3)$ Å$^3$ for NFPP-E compound. The typical crystalline structure is presented in the inset of Fig. 1a. In the open 3D robust framework, and the $[Fe_3P_2O_{13}]$ units along the $a$-axis each consist of three $FeO_6$ octahedra and three $PO_4$ groups, while the $[Fe_3P_2O_{13}]_\infty$ infinite layers are connected by $[P_2O_7]$ groups along the $a$ direction, thus leading to the formation of large primary tunnels along the $b$ direction. The NFPP-C sample also shows good crystallinity with the space group $Pn2_1a$ (as shown in Supplementary Fig. 1). We summarize the detailed atomic site information for both the NFPP-E and the NFPP-C samples in Supplementary Table 1 and Supplementary Table 2. We also observed trace amounts of maricite ($NaFePO_4$) impurities (around 4% in both NFPP-E and NFPP-C samples). Supplementary Fig. 2 shows the paramagnetic properties (magnetization curve (M-H) and temperature dependence of the magnetic susceptibility (M-T)) for both samples from probing the spin state of Fe. It can be seen that the hysteresis loop of NFPP-C is slightly larger than that of NFPP-E, which can be ascribed to its slightly higher proportion of impurities, as indicated by the refinement. The calculated effective magnetic moments ($\mu_{eff}$) of NFPP-E and NFPP-C are 5.04 $\mu_B$/Fe and 5.19 $\mu_B$/Fe per formula unit, respectively, which are close to the

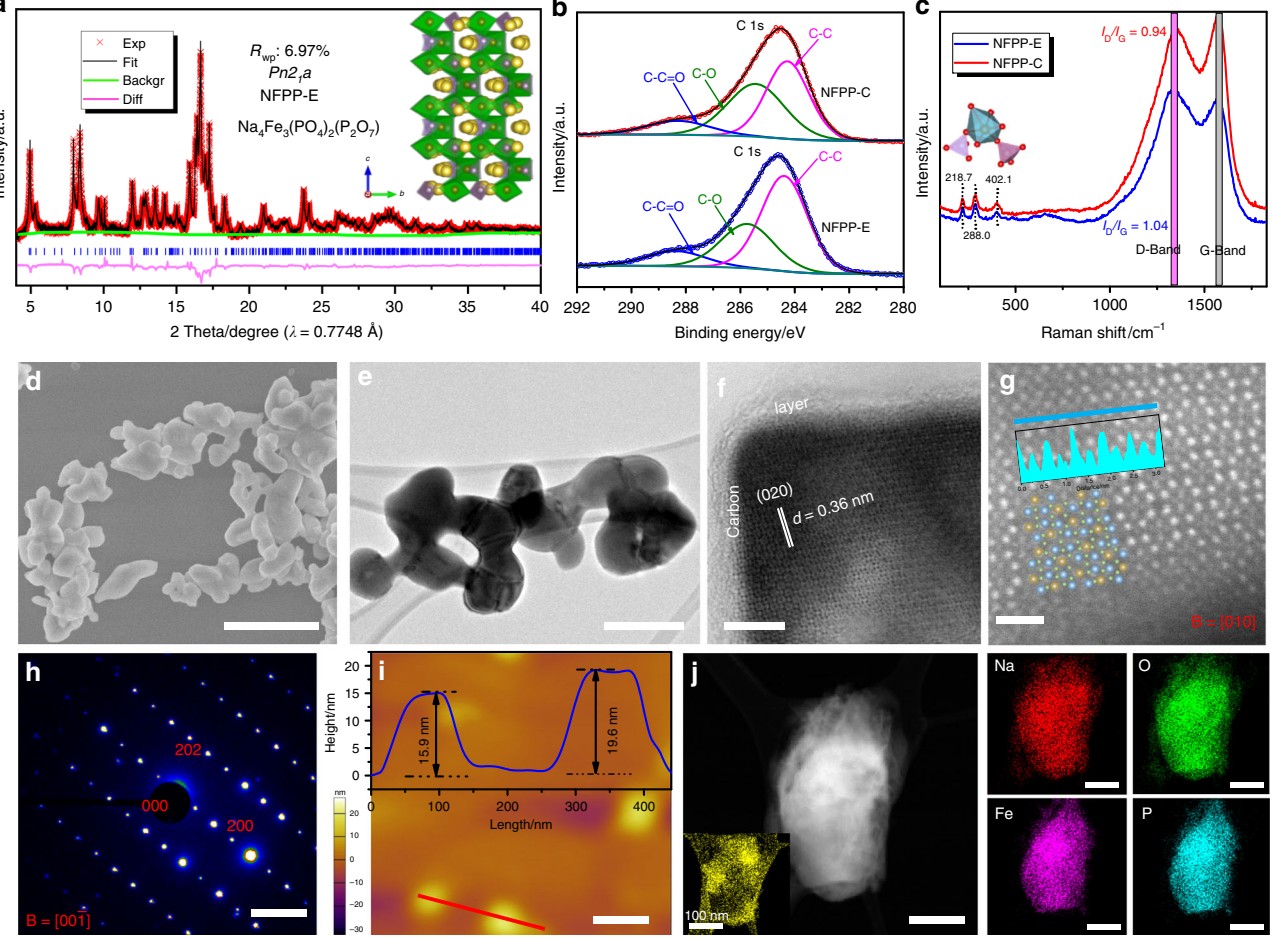

**Fig. 1** Characterizations of as-prepared sample. Rietveld refinements of **a** NFPP-E. Schematic representation of the refinement results is presented in the insets. **b** XPS spectra of C 1s and the corresponding deconvolution curves of both NFPP-E and NFPP-C samples. **c** Raman spectra of both samples in the Raman shift range from 100 cm$^{-1}$ to 1800 cm$^{-1}$. **d** SEM images of NFPP-E and **e** transmission electron microscope (TEM) image of NFPP-E particles. **f** Bright field (BF) image of NFPP-E with carbon layers. **g** HAADF image from aberration-corrected STEM and the crystal structure of NFPP-E viewed from the [010] direction. The insets are the corresponding signal responses along the selected lines. **h** The corresponding SAED pattern of NFPP-E and **i** atomic force microscopy (AFM) image of NFPP-E nanoplate particles and their corresponding heights. **j** The STEM-EDS mapping results for selected elements of NFPP-E. Scale bars: 500 nm (**d**); 200 nm (**e**); 5 nm (**f**); 1 nm (**g**); 2 1/nm (**h**); 200 nm (**i**); 100 nm (**j**)

theoretical value (4.89 μ$_B$/Fe per formula unit) for high-spin Fe$^{2+}$ ($d^6$, $t_{2g}^4$ $e_g^2$, $S = 2$)[33].

The carbon contents of NFPP-E and NFPP-C were detected to be 3.6 wt. % and 4.1 wt. %, respectively, according to the thermogravimetric (TG) analysis in Supplementary Fig. 3. We further characterized the carbon via employing X-ray photoelectron spectroscopy (XPS) and Raman spectroscopy. Peak deconvolutions were carried out to fit the C 1s spectra to three main bonding configurations (Fig. 1b). The typical π-bonding graphite-like carbon peaks for delocalized electrons are located at 284.6 eV in both samples. Peaks with higher energies ($sp^3$ carbon), namely, C-O (285.8 eV) and O=C-O (288.7 eV) were observed and fitted[13]. It was found that 65.9% of the carbon was graphitized carbon in the NFPP-E sample, while this value in NFPP-C was only 52.3%, which indicates that the carbon conductivity of NFPP-E may higher than that of NFPP-C. In addition, the Fe 2p XPS spectra in Supplementary Fig. 4 reveal the chemical compositions and surface electronic states of Fe element. No obvious difference can be detected except for a tiny binding energy shift, and the Fe deconvolution curves indicate that the valence of Fe in both samples is similar. The Raman spectra of Na$_4$Fe$_3$(PO$_4$)$_2$(P$_2$O$_7$) range from 150 to 800 cm$^{-1}$, as shown in

Fig. 1c. The detected peaks at 218.7 cm$^{-1}$ and 288.0 cm$^{-1}$ can be assigned to the stretching vibrations and bending motions of the PO$_4$ tetrahedra, respectively, and the peak located at 402.1 cm$^{-1}$ is mainly due to the stretching vibrations of FeO$_6$ octahedra[34]. It is worth noting that the intensity ratio of the D band to the G band ($I_D/I_G$) for the NFPP-E sample (1.04) is larger than that for the NFPP-C sample (0.94). The D-band is mainly due to defects, edges, or structural disorder while the G-band normally represents the $E_{2g}$ mode of $sp^2$ carbon layers. This phenomenon indicates that the carbon conductivity of the NFPP-E particle surface would be higher than that of NFPP-C, which is in good accordance the C 1 s deconvolution curves in Fig. 1b. The typical vibration modes of both the PO$_4$ units and the FeO$_6$ octahedra were also detected in the Fourier transform infrared (FT-IR) spectrum (Supplementary Fig. 5). The predominant overlapped peak of the $\nu_1$ and $\nu_3$ symmetric and asymmetric stretching modes of the PO$_4$ units is around 1109 cm$^{-1}$, while the $\nu_2$ and $\nu_4$ symmetric and asymmetric bending modes of the PO$_4$ tetrahedra are responsible for the splitting of the O-P-O peaks in the range from 590 to 740 cm$^{-1}$. The peak around 543 cm$^{-1}$ represents the vibration of the bonds between Fe$^{2+}$ and O$^{2-}$ in the isolated FeO$_6$ octahedra[19]. The presence of O-C=O (2361 cm$^{-1}$)

asymmetric stretching and C=O vibrations indicates the nature of the carbon bonded to the surface of the as-obtained materials.

The typical scanning electron microscope (SEM) and scanning transmission electron microscope (STEM) images in Fig. 1 and Supplementary Fig. 6 visually confirm the nanosized and microporous morphologies of the as-prepared NFPP-E and NFPP-C samples. The NFPP-E particles feature a nanoplate-like morphology with an average particle size around 150 nm, as shown in Fig. 1d and e, while the NFPP-C particles crystallized in an average particle size of 2 μm with abundant microporous architecture on the surface, as displayed in Supplementary Fig. 6a–c. We employed Brunauer–Emmett–Teller (BET) testing to detect the specific surface areas, and values of 3.52 and 9.74 $m^2 g^{-1}$ were obtained for the NFPP-E and NFPP-C samples (Supplementary Fig. 7a), respectively. The pore size distributions are displayed in Supplementary Fig. 7b, and it was found that the average pore size of the NFPP-C particles is slightly larger than that of the NFPP-E, which in good accordance with their porous nature, as observed in the SEM images. Both the samples displayed good single crystallinity with the indexed space group $Pn2_1a$, and the coated carbon layers on the NFPP-E and NFPP-C particles (Fig. 1f and Supplementary Fig. 6d) were measured to be 4 nm and 3 nm in thickness, respectively. Typical high-angle angular dark-field (HAADF) images exhibit an atomic-level crystal structure viewed along specific crystallographic directions, where the positions of Fe and P heavy atomic columns can be clearly identified (Fig. 1g and Supplementary Fig. 6e). Since the signal intensity in a HAADF image is proportional to the atomic number ($Z$), the O atoms are invisible. The relative positions and contrast of Fe, P, and Na can be better acquired with the help of linear profiles. Framework structures with void space along the [010] and [111] directions were identified for the NFPP-E and NFPP-C samples, respectively, and the corresponding schematic illustrations are also presented as insets. Selected area electron diffraction (SAED) patterns of NFPP-E and NFPP-C samples also confirmed the well-crystallized structure of both samples (Fig. 1h and Supplementary Fig. 6f). Atomic force microscopy (AFM) was used to measure the heights of the nanoplates, and an average height of 18 nm for the NFPP-E nanoparticles was observed (Fig. 1i). Energy dispersive spectroscopy (EDS) mapping results were also acquired via STEM-EDS, and the results are displayed in Fig. 1j and Supplementary Fig. 6g. The mapping results indicate that the elements Na, Fe, P, O, and C coexist and are distributed uniformly in both the NFPP-E and the NFPP-C particles, which is in good agreement with the refined powder diffraction results.

**Electrochemical properties investigations of $Na_4Fe_3(PO_4)_2(P_2O_7)$ materials**. The electrochemical properties of both the NFPP-E and the NFPP-C samples were tested in coin cells with sodium metal anodes. The loading density of active material for each prepared electrode was measured to be about 2.0 mg cm$^{-2}$ to avoid weighting deviations. The electrolyte consisted of ethylene carbonate/propylene carbonate (EC/PC, 1:1 by volume) with 1 M NaClO$_4$ as the sodium salt and 5 vol. % fluoroethylene carbonate (FEC) as additive[35]. The NFPP-E electrode showed excellent electrochemical performance at various current densities (Fig. 2a). It delivered discharge capacities of 113.0 mAh g$^{-1}$ and 108.3 mAh g$^{-1}$ at 0.05 C and 0.1 C (1 C = 120 mA g$^{-1}$), and even up to 20 C, there was still 80.3 mAh g$^{-1}$ remaining, which is completely comparable with the known sodium superionic conductor (NASICON)-structured polyanionic materials, such as the well-studied Na$_3$V$_2$(PO$_4$)$_3$ material[13]. We also compared the C-rate performances of recently published iron-based polyanionic materials, and it can be found that our as-obtained material

shows the best C-rate performance among them up to 20 C (Supplementary Fig. 8)[19,29,34,36–46]. At small current densities, there was no obvious discrepancy between the NFPP-E and NFPP-C electrodes, indicating the high ionic conductivity of the Na$_4$Fe$_3$(PO$_4$)$_2$(P$_2$O$_7$) particles. The inset of Fig. 2a presents the cycling stability of both electrodes at 0.05 C for 50 cycles, and there was almost 100 % retention for both samples, reflecting the robust framework of this mixed-polyanion system. Also, the initial Coulombic efficiency (ICE) was nearly 100% for both samples, which will greatly facilitate their real application in full SIBs. The high rate performance of NFPP-E was better than that of NFPP-C, which can be explained by the nanosized particles, which shortened the sodium diffusion distances. Excellent reversibility was obtained with no obvious discrepancy between the first and second cycles except for small changes in peak position due to solid electrolyte interphase (SEI) layer formation (Fig. 2b), which indicates that topotactic single phase variation takes place during cycling without iron atom migration to their favoured face-sharing tetrahedral sites, which may result in a non-equilibrium phase transition (usually seen in alluaudite or pyrophosphate frameworks). The same situation between the first two cycles of NFPP-C electrode was observed and is shown in Supplementary Fig. 9a. Also, the reduced charge transfer resistance after the first cycle in the electrochemical impedance spectroscopy (EIS) spectra indicates the formation of a SEI layer on the particle surface (Supplementary Fig. 9b, c). Much alleviated electrochemical polarization compared with that of NFPP-C can be observed the NFPP-E electrode in Fig. 2c. In addition, the as-prepared NFPP-E electrode showed outstanding cycling stability at various C-rates. Figure 2d displays the cycling performance at 0.2 C and 0.5 C for 250 cycles and 430 cycles, respectively, and capacity retention of 85.0% and 84.0% could be obtained. It is worth noting that there was almost no mid-voltage decay (retention of 98.9%) within the first 430 cycles (Supplementary Fig. 10a), and this value is very important in terms of the total energy density for practical use. We also tested both NFPP-E and NFPP-C electrodes at high rate (20 C, Fig. 2e), and similar capacity degradation was obtained for both electrodes, indicating that the morphology and particle size are the only reasons for the electrochemical discrepancy. Capacity retention of 69.1% and 57.2% for NFPP-E and NFPP-C, respectively, was achieved after 4400 cycles. Then, we extracted the electrodes from the coin cells to acquire more details. No obvious cracks can be seen on the surfaces of the NFPP-E and NFPP-C electrodes (Supplementary Fig. 10b–e). From Supplementary Fig. 11, it can be seen that the primary morphologies were not destroyed even after 4400 high-rate cycles. Good crystallinity still could be founded in the high-resolution TEM (HRTEM) images as well as the SAED patterns of both samples. Supplementary Fig. 12 shows the STEM-EDS mapping results after 4400 cycles. All the existing elements (Na, P, Fe, O, C, and F (from the binder)) were uniformly distributed, and the porous structure of NFPP-C was well maintained.

Galvanostatic intermittent titration technique GITT testing was carried out in a coin cell after it was given 30 cycles to reach its thermal equilibrium state (Fig. 2f). It can be seen that a single phase (solid-solution) reaction mechanism appeared in both the charge and the discharge processes. The theoretical capacity (128.9 mAh g$^{-1}$) is based on a triple electron transfer process, so the sodium diffusion coefficients fluctuated during the continuous sodium de-/insertion from/into the framework (inset image in Fig. 2f). The sodium diffusion coefficients were in the range from $10^{-13}$ to $10^{-10}$ cm$^2$ s$^{-1}$ within the voltage window of 2.7–4.1 V, a performance which is highly comparable to those recognized NASICON-type cathode materials with similar orders of magnitude. We summarize the details of the calculation in Supplementary Fig. 13 and Supplementary Note 1. In addition,

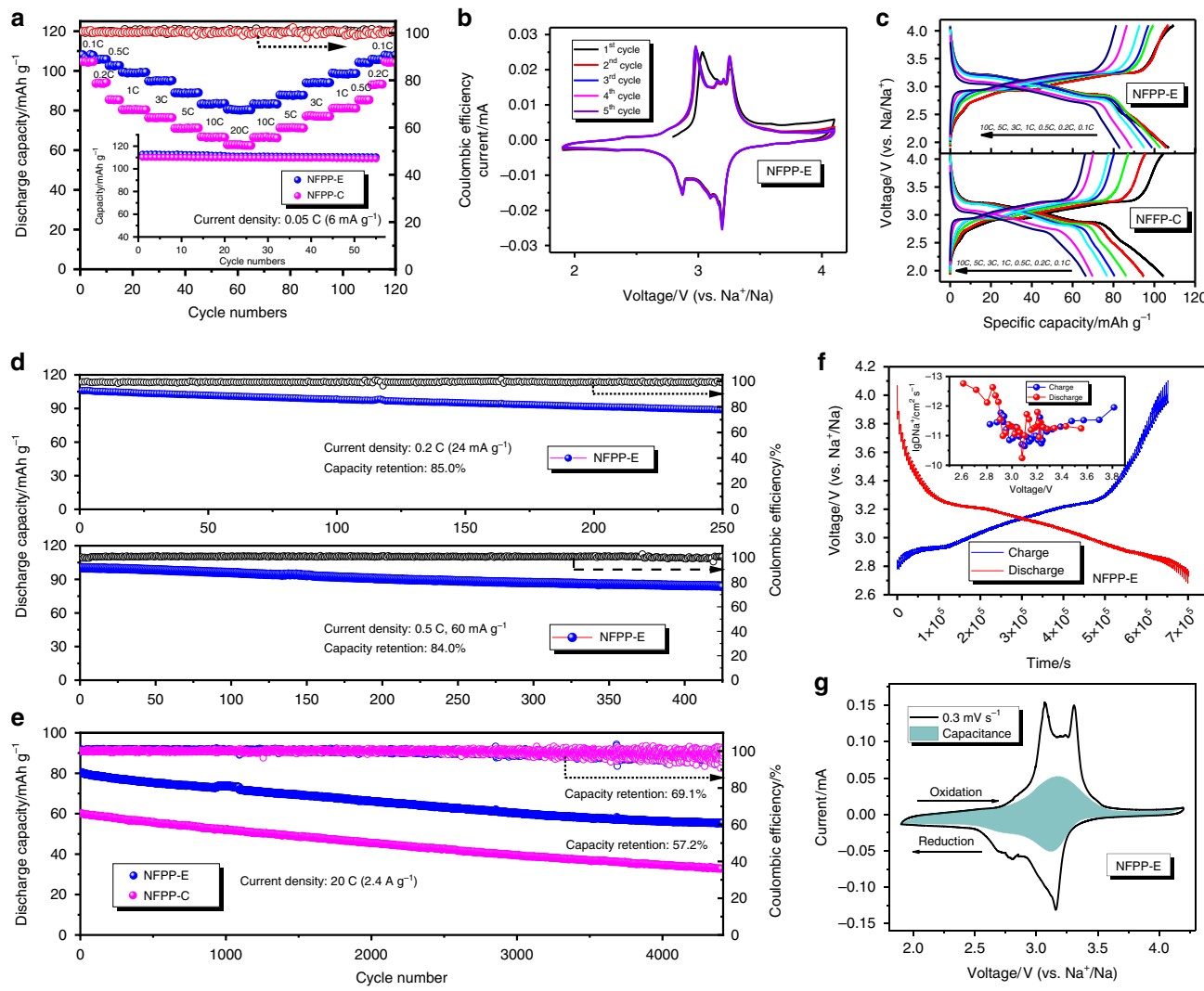

**Fig. 2** Electrochemical profiles of both samples. **a** Rate capability of both samples from 0.1 C to 20 C. The inset is the cycling stability at low current density (0.05 C). **b** Cyclic voltammetry (CV) curves for the first 5 cycles of NFPP-E electrode (scan rate 0.05 mV s$^{-1}$). **c** Charge-discharge curves at different rates for both samples. **d** Cycling stability of NFPP-E electrodes over 250 cycles at 0.2 C and 430 cycles at 0.5 C. **e** Long-term cycling stability (4400 cycles) at high rate (20 C) for both NFPP-E and NFPP-C electrodes. **f** Galvanostatic intermittent titration technique (GITT) curves of NFPP-E material for both charge and discharge processes. The inset is the chemical diffusion coefficient of Na$^+$ ions as a function of voltage calculated from the GITT profile (after 30 cycles, current density: 0.05 C). **g** The calculated capacitance contribution (shadowed area) to the CV curve of NFPP-E at the scan rate of 0.3 mV s$^{-1}$

the capacitance was determined from various scan rates (Supplementary Fig. 14 and Supplementary Note 2, from 0.05 to 0.4 mV s$^{-1}$). It is believed that non-faradic and faradic processes always coexist in the charge storage mechanism. The faradic process can provide a fixed working potential, while the non-faradic process (commonly regarded as pseudocapacitance) can help by providing fast charge transitions with extended cycling life. The *b* value was determined to be around 0.72 with four redox peaks identified, and the non-faradic process was calculated to be responsible for 23.4% of the total current at 0.3 mV s$^{-1}$ (Fig. 2g). This can be regarded as one of the key factors behind the outstanding electrochemical properties of NFPP-E electrode.

**Air stability, all-climate, and full cell performances**. We further tested the air stability and all-climate performance of NFPP-E material to provide more practical details relevant to its real applications. Figure 3a shows an XRD comparison between the fresh powder and the powder exposed to air for three months. No

obvious peak shifts or variations can be seen. In addition, from the XPS results in Fig. 3b, there are no detectable discrepancies or peak shifts that can be observed. We collected EIS spectra and found that only a negligible difference appeared in the charge transfer resistance, which might be ascribed to the individual cell assembly processes (Fig. 3c). The morphologies remained unchanged (Supplementary Fig. 15), and the particles surfaces were still well crystallized even after being in contact with air for three months (Fig. 3d). From Fig. 3e and f, it is clear that almost the same electrochemical performance can be achieved for the NFPP-E electrodes under all the various conditions, and the Coulombic efficiency was maintained at around 100% for each cycle. From the characterizations conducted above, it can be deduced that this Na$_4$Fe$_3$(PO$_4$)$_2$(P$_2$O$_7$) mixed polyanionic material has an air stable nature, which is quite different from the case of the pyrophosphates, which are usually unstable in air and face electrochemical degeneration[39,47]. Thus, this material is also very promising for aqueous SIB systems[48]. We assume that both the carbon coating layer and the robust phosphate anionic groups contribute to the air stability of this Na$_4$Fe$_3$(PO$_4$)$_2$(P$_2$O$_7$)

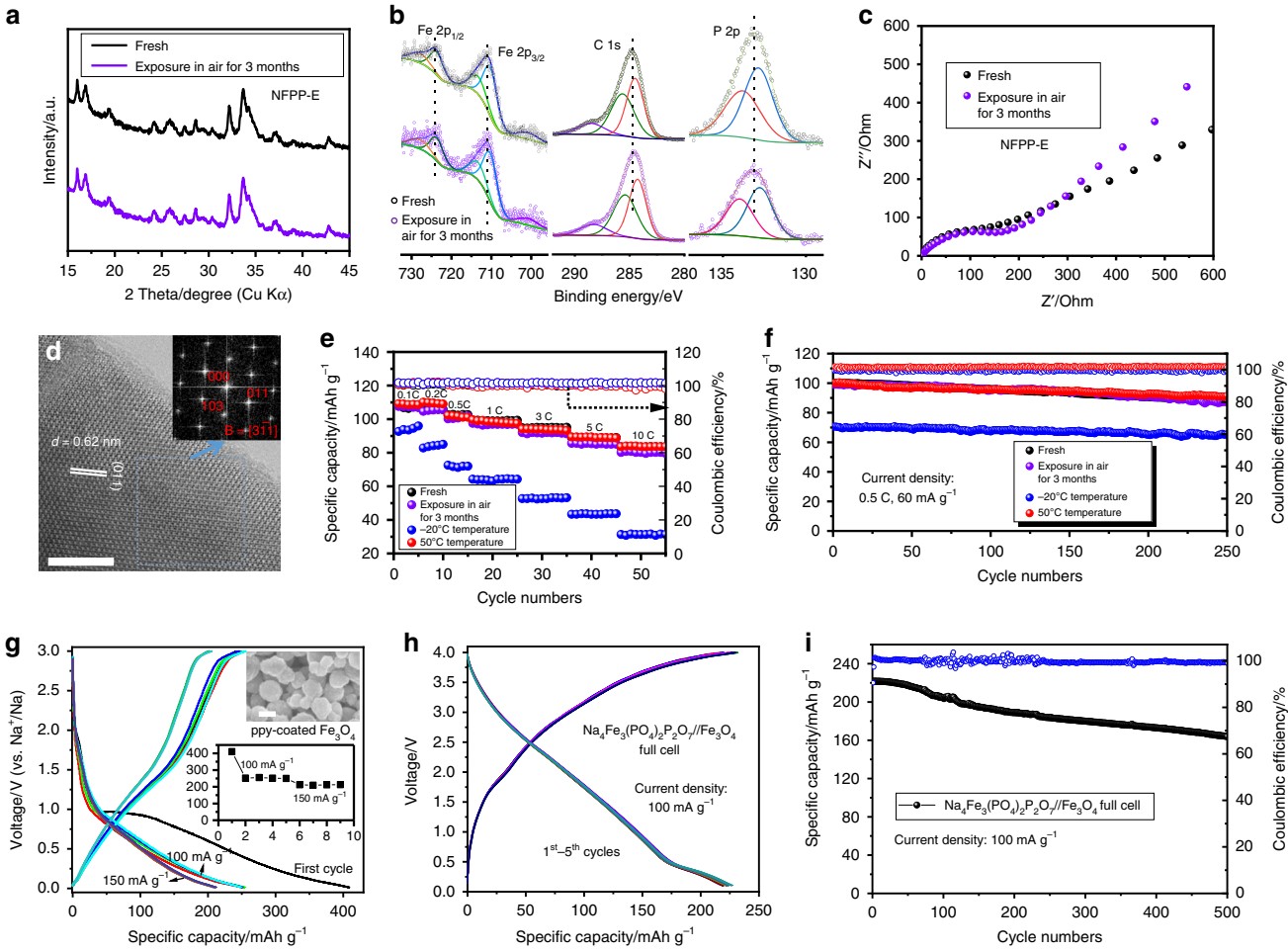

**Fig. 3** Air stability, low/high-temperature performance, and full cell performance. **a** XRD comparison of NFPP-E powder in the fresh state and after exposure to air for three months (Cu Kα radiation, λ = 1.5406 Å). **b** Fe 2p, C 1s, and P 2p XPS fitted spectra of NFPP-E powder in the fresh state and after exposure to air for three months. **c** EIS spectra of NFPP-E powder in the fresh state and after exposure to air for three months. **d** HRTEM image of the powder after exposure to air for three months. The inset is the fast Fourier transform (FFT) pattern of the marked area. **e** C-rate capability of NFPP-E electrodes: fresh, exposed to air for three months, and fresh at both −20 °C and 50 °C. **f** Cycling performances of the four electrodes in **e**. **g** Charge/discharge curves of polypyrrole (PPy)-coated $Fe_3O_4$ at various current densities. The insets are (top) SEM image of as-prepared $Fe_3O_4$ nanospheres and (bottom) the cycling performance. **h** Voltage profiles of the all-iron-based $Na_4Fe_3(PO_4)_2(P_2O_7)$//$Fe_3O_4$ full cell for the first five cycles. **i** Cycling stability of the $Na_4Fe_3(PO_4)_2(P_2O_7)$//$Fe_3O_4$ full cell. The specific capacities of the full cell were all based on the anodes. Scale bars: 10 nm (**d**); 200 nm (**g**, inset)

material. In addition, testing of its all-climate performance was carried out at both −20 °C and 50 °C. It can be seen that at low C-rates at −20 °C, the specific capacity of NFPP-E could reach 95.0 mAh g$^{-1}$ and 84.7 mAh g$^{-1}$ at 0.1 C and 0.2 C, respectively, but a fast capacity drop was encountered with increased current density. There was almost no difference between room temperature and 50 °C for the NFPP-E electrodes, however, indicating that their superior electrochemical performance can be well maintained in hot climate regions. The charge/discharge curves for temperature comparison are displayed in Supplementary Fig. 16. The cycling stability of NFPP-E electrode at −20 °C and 50 °C remained outstanding, with 92.1% and 91.4% capacity retention at 0.5 C, respectively, indicating that neither low nor high temperatures had any further influence on the crystal structure apart from the kinetics factors. In addition, we also fabricated polypyrrole (PPy)-coated $Fe_3O_4$ nanospheres to make all-iron-based low-cost SIB full cells. The preparation details for the anodes can be found in the experimental section. Supplementary Fig. 17 shows the XRD pattern and morphology of the as-obtained PPy-coated $Fe_3O_4$ nanospheres. Figure 3g displays the electrochemical behaviour of the as-obtained PPy-coated

$Fe_3O_4$ nanospheres, and capacity of around 250 mAh g$^{-1}$ and 210 mAh g$^{-1}$ was achieved and stabilized at 100 mA g$^{-1}$ and 150 mA g$^{-1}$, respectively. As shown in Fig. 3h, an all-iron-based low-cost SIB full cell was activated and operated at 100 mA g$^{-1}$ in the voltage window of 0.1–4.0 V. It should be pointed out that, due to the low ICE of $Fe_3O_4$, the anodes in the full cells were precycled. The first cycle reversible efficiency reached as high as 93.1 %, and no obvious curve discrepancies can be observed, indicating the excellent reversibility of the as-fabricated full cell. All the specific capacities of the full cell are based on the anodes. In Fig. 3i, the full cell achieves capacity retention of 76.9% at 100 mA g$^{-1}$ over 500 cycles and high Coulombic efficiency near 100%. In addition, with more practical concerns, we also fabricated the full cells with commercial hard carbon as anode (purchased from KURARAY Co., Ltd., Japan (Type 2)). SEM images and the electrochemical performance at 100 mA g$^{-1}$ are displayed in Supplementary Fig. 18a–c. We then fabricated a $Na_4Fe_3(PO_4)_2(P_2O_7)$//Hard carbon full cell with the loading mass ratio of 1.8:1 for capacity balance according to the individual specific capacities of the electrodes. The anode electrodes were presodiated to reduce the dramatic initial irreversible capacity loss.

The results are shown in Supplementary Fig. 18d, e. It can be seen that the charge/discharge curves of the $Na_4Fe_3(PO_4)_2(P_2O_7)$// Hard carbon full cell are not as sloping as for those using PPy-coated $Fe_3O_4$, so the energy density can be improved with an elevated mid-working voltage platform. Nevertheless, the capacity is continuously dropping within the initial 10 cycles. We cycled the fabricated full cell for up to 120 cycles at $100\,mA\,g^{-1}$ and found that the capacity retention is less than 50 %. The charge capacity was always higher than discharge capacity for every cycle, which might be the main reason for the continuous capacity drop. The Coulombic efficiency of commercial hard carbon (97% each cycle) is likely to be another reason for this. Nevertheless, we are still putting much effort into working towards better full cell performance using commercial hard carbon as anode with an optimised electrolyte system, more suitable loading ratio, etc. to be used in our future work.

**Sodium-storage mechanism.** In order to obtain a more comprehensive understanding of the structural superiority of $Na_4Fe_3(PO_4)_2(P_2O_7)$ material, both in-situ synchrotron-based X-ray diffraction patterns and X-ray absorption spectra (XAS) were collected on the DESY beamline, Germany. The wavelength was changed to 1.5406 Å (Cu Kα) for better comparison. A two-dimensional (2D) XRD pattern and the original in-situ pattern without wavelength change are displayed in Supplementary Fig. 19. Several peaks in the initial states were indexed and identified, as shown in Fig. 4a, b. It can be seen that all the indexed peaks changed reversibly during the charge/discharge process with the general patterns remaining unchanged, indicating that the robust crystal framework can be well maintained after electrochemical activation. Major peaks such as (200), (011), (210), and (222) gradually moved to higher 2θ values during sodium insertion and returned to their original values during sodium de-insertion, which can be attributed to the continuous lattice volume variations during cycling. No asymmetric variations were observed, indicating that there were no crystal distortions or cation migration during sodium de-/insertion. Therefore, it can be deduced that a topotactic one-phase transition occurred in the $Na_4Fe_3(PO_4)_2(P_2O_7)$ electrode during cycling. We also obtained HRTEM images of the fully charged and fully discharged NFPP-E electrode (Fig. 4c, d). Lattice fringes for the (200) and (202) planes are clearly observed and identified, indicating the high structural crystallinity of the NFPP-E material during all of the electrochemical reactions. The remaining $Na^+$ ions in the crystal structure (around 25 %) can be regarded as belonging to the binding pillars. In Fig. 4e, f, the cell parameters were calculated from selected peaks during the first cycle with reversible lattice breathing. The corresponding volume change was calculated to be only 4.0% during the first charge process. Such a small volume change in the crystal structure guarantees long-term cycling stability, as demonstrated. Furthermore, the valence state of Fe in $Na_4Fe_3(PO_4)_2(P_2O_7)$ during electrochemical reactions was evaluated using in-situ X-ray absorption near-edge structure (XANES) analysis, as shown in Fig. 4g–i. We employed $FePO_4$ and $LiFePO_4$ as the references for the valences of $Fe^{2+}$ and $Fe^{3+}$, respectively. The 2D contour plot (Fig. 4g) of the normalized XANES spectra clearly shows the reversible variation in the valence of Fe in the X-ray energy range around 7120 eV. Figure 4h also shows that the XANES spectrum is shifted to the right (higher energy area) during charging, indicating oxidation of iron from $Fe^{2+}$ to $Fe^{3+}$. The spectrum of the fully charged state was very close to that of the $FePO_4$ reference sample, indicating that most of the iron in the crystal structure had become $Fe^{3+}$. The pre-edge of Fe K-edge during charging also showed discernible variation (inset in Fig. 4g), with the peak shifted towards

higher energy. Then, the XANES spectrum moved back to the lower energy area during the discharge process (Fig. 4i), and the corresponding pre-edge spectra shifted accordingly (inset in Fig. 4i). It should be pointed out that the spectra of both the initial state and the discharge state were almost the same, but with obvious discrepancy compared to the $LiFePO_4$ reference sample. This can be ascribed to the individual fingerprint information on the $P_2O_7$ and $PO_4$ groups. Also, the possibility of oxidation of the remaining $Fe^{2+}$ in the crystal structure during cycling still cannot be neglected. On the basis of the in-situ XRD and in-situ XANES analyses, we can confirm that both the robust reversible crystal framework and the continuous valence change of Fe contributed to the excellent electrochemical performance shown above.

**Na-ion dynamics.** In order to deeply understand the intrinsic properties of the crystal structure that are responsible for the outstanding C-rate performance, both BVS and DFT calculations were employed to investigate the localized migration energy barriers. The BVS calculation is a well-established empirical tool for preliminary examination of ionic states and diffusion pathways. Both the bond-valence map and the bond-valence electron voltage map are shown in Supplementary Fig. 20. The isosurface value was set as 1 where the $Na^+$ ions can possibly be found (Supplementary Fig. 20a, b), and the isosurface near $-3.5\,eV$ shows the lowest initial energy regions, indicating possible $Na^+$ ion diffusion pathways (Supplementary Fig. 20c, d). The corresponding 2D slice images reveal details of the lowest energy regions in various directions (Supplementary Fig. 21). We found that along the $a$, $b$, and $c$ axes, the lowest energy regions are well connected, which means that this material possibly possesses more than one-dimensional $Na^+$ ion diffusion pathways. Therefore, we conducted a DFT study to further determine the details. We discovered that all the $Na^+$ ions in a single unit can be assigned to three different types, based on their individual binding energies. The detailed information, as well as calculation method, is summarised in Supplementary Table 3. In Fig. 5, various images of the crystal structure of $Na_4Fe_3(PO_4)_2(P_2O_7)$ material with three different types of $Na^+$ ions are presented. It was calculated that the diffusion energy barriers within the same $Na^+$ ion type were 0.553 eV, 0.02 eV, and 0.365 eV for A to A type, B to B type, and C to C type, respectively, which are all very low energy barriers for the transfer of $Na^+$ ions (Fig. 5b). In particular, it can be concluded that the most favourable diffusion tunnel in the first stage is along the $a$ direction, since almost no energy barrier (0.02 eV) was detected. Then, we performed the calculations between different $Na^+$ ion types, which are equivalent to three-dimensional diffusion pathways, since $a$, $b$, and $c$ directions are all involved. From Fig. 5e, it can be seen that all the energy barriers for $Na^+$ ion diffusion are lower than 0.9 eV, which are all possible diffusion pathways in the presented crystal structure, providing solid evidence for this newly recognized NASICON-type structure with 3D diffusion pathways or tunnels in this $Na_4Fe_3(PO_4)_2(P_2O_7)$ material. The NASICON framework is constructed from corner-sharing $MO_6F_{6-x}$ (M=Ti, V, etc.) octahedra and $XO_4$ (X = P, S, Si, etc.) tetrahedra. The $XO_4$ tetrahedra maintain the charge balance and interact with the bipolar $MO_6F_{6-x}$ octahedra, while the $MO_6F_{6-x}$ octahedra can offer the main electrostatic repulsion that accounts for the various voltage platforms for $Na^+$ intercalation/extraction. The critical factor for a NASICON-type structure is the arrangement of sodium sites and whether the 3D diffusion pathways of $Na^+$ can be accessed with relatively low energy barriers. We also performed ionic conductivity testing, and it was found that the total conductivity in various temperatures of the $Na_4Fe_3(PO_4)_2(P_2O_7)$ material was of the same order of magnitude compared with the

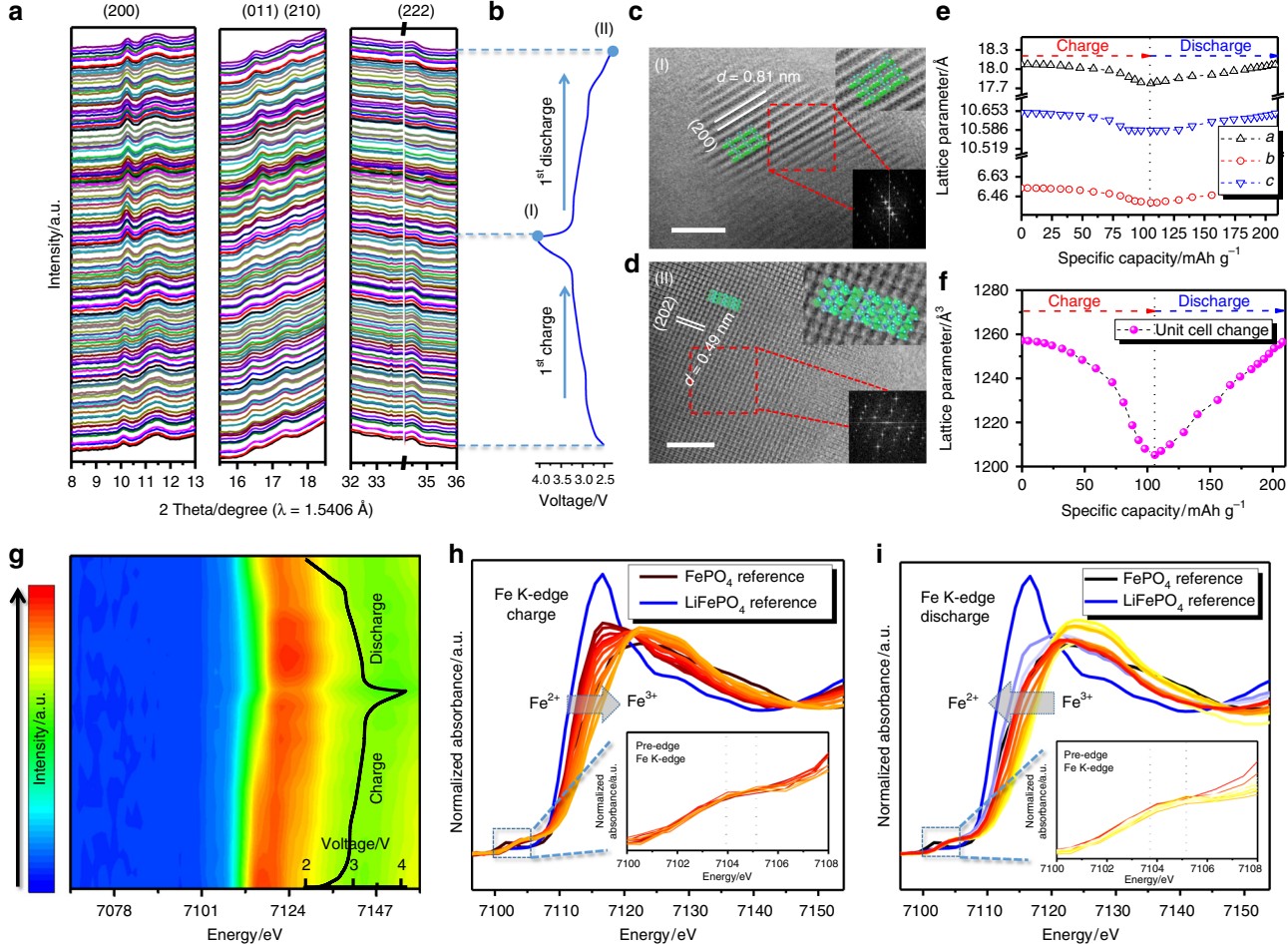

**Fig. 4** Sodium-storage mechanism investigations. **a** In-situ synchrotron-based XRD patterns and **b** corresponding charge-discharge curves of NFPP-E electrode during the first cycle. **c**, **d** HRTEM images of fully charged and fully discharged NFPP-E electrodes. **e** Variations of *a*, *b*, and *c* lattice parameters during the charge/discharge process. **f** Volume change details during the charge/discharge process. **g** In-situ XANES spectra at the Fe K-edge of NFPP-E electrode (2D contour plot) as a function of charge-discharge curve. **h** Charge process and **i** discharge process of typical Fe K-edge XANES spectra. The insets in **h** and **i** are the corresponding pre-edge spectra. Scale bars: 5 nm (**c**); 5 nm (**d**).

well-recognised NASICON-type $Na_3V_2(PO_4)_3$ electrode (Supplementary Fig. 22)[49]. Both the BVS and the DFT calculations revealed the intrinsic reasons for the outstanding high rate performance of this material, which are completely comparable to those of the known NASICON-type materials mentioned above. Considering the long-term cycling stability, high-rate capability, air stability, and all-climate performance of this low-cost mixed polyanionic material, we believe that this new NACISON-type $Na_4Fe_3(PO_4)_2(P_2O_7)$ material is a strong competitor among the various sodium hosts competing for real applications in large-scale ESSs[50–54].

## Discussion

We have successfully synthesized tuneable nanosized $Na_4Fe_3(PO_4)_2(P_2O_7)$ plates and microporous $Na_4Fe_3(PO_4)_2(P_2O_7)$ particles via a facile one-step sol-gel method with high phase purity and uniform carbon coating. As cathode, excellent rate performances of 113.0 and 80.3 mAh g$^{-1}$ at 0.05 C and 20 C, respectively, were achieved, and impressive cycling stability was obtained without noticeable voltage decay (69.1% capacity retention after 4400 cycles at 20 C) with almost 100% ICE achieved. The air stability and all-climate (−20 °C and 50 °C) performance were investigated, and this material showed

outstanding air stability and all-climate electrochemical properties (84.7 mAh g$^{-1}$ at 0.2 C) at −20 °C. A low-cost all-iron-based full cell was fabricated and characterized with a view towards real applications. Both the in-situ XRD and the in-situ XANES results revealed its excellent crystal reversibility and structural stability during cycling. A GITT study showed high sodium diffusion coefficients, and its NASICON-type structure and 3D diffusion pathways were further explored and verified by both BVS and DFT calculations in detail, revealing impressive low activation energy barriers. Our studies suggest that this low-cost mixed polyanionic $Na_4Fe_3(PO_4)_2(P_2O_7)$ material is a strong competitor among the various sodium hosts and should receive more comprehensive investigation towards its real application in large-scale energy storage systems in the near future.

## Methods

**Synthesis of $Na_4Fe_3(PO_4)_2(P_2O_7)$ particles**. Both the nanosized $Na_4Fe_3(PO_4)_2(P_2O_7)$ plates (NFPP-E) and the microporous $Na_4Fe_3(PO_4)_2(P_2O_7)$ particles (NFPP-C) were prepared via typical sol-gel methods. All the chemicals used in this paper were analytically pure, and purchased from SIGMA-ALDRICH without further purifications. In a typical process, 4 mmol sodium acetate, 4 mmol ammonium phosphate, 0.1 g glucose, and 0.1 g stearic acid were added to 20 ml deionized water with magnetic stirring until a transparent solution was obtained (denoted as solution A). 3 mmol iron (II) acetate, 0.8768 g ethylenediaminetetraacetic acid, and 0.05 g cetyltrimethylammonium bromide were added to 20 ml

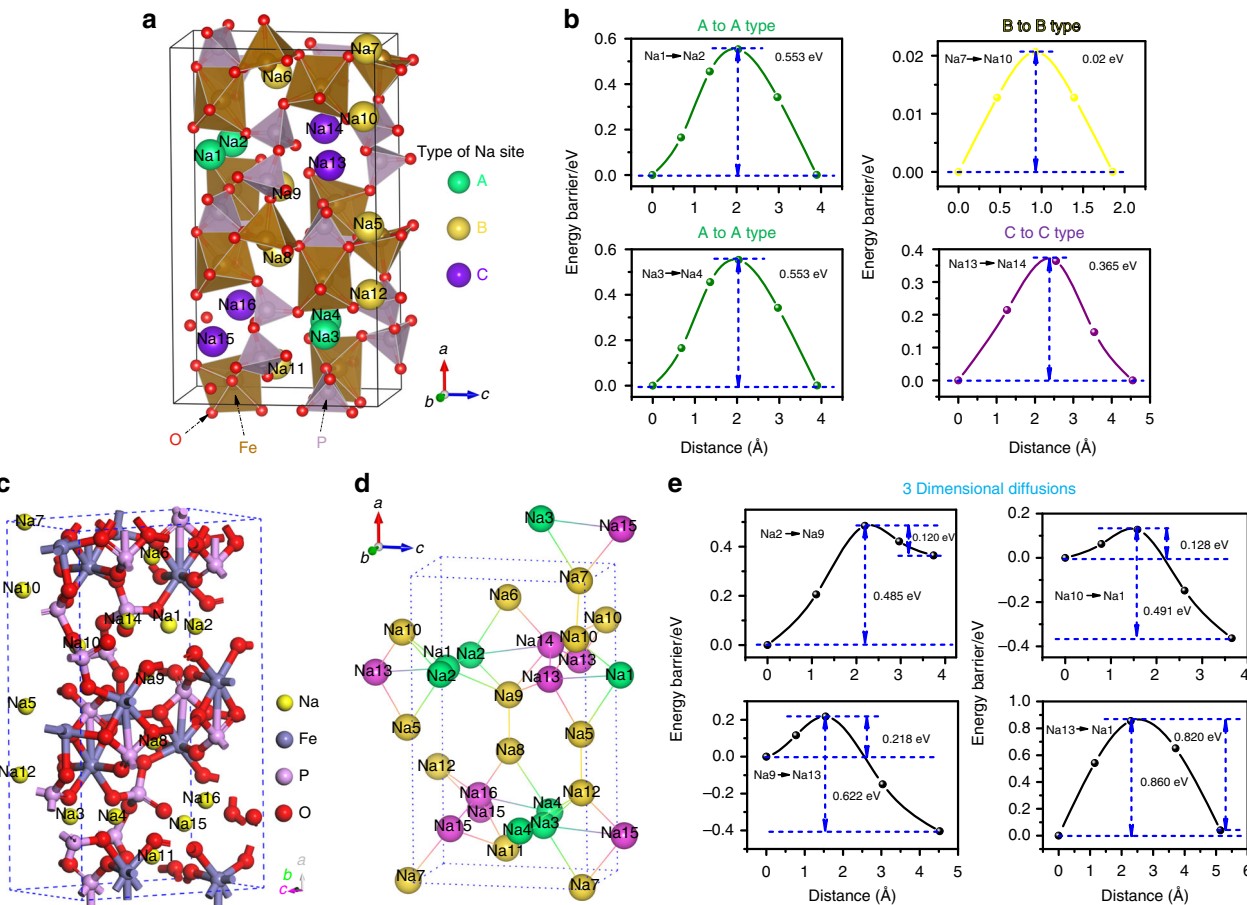

**Fig. 5** Crystal structure, Na$^+$ ion diffusion paths, different types of Na$^+$ ions in a single unit, and corresponding migration energy barriers. **a**, **c**, **d** Various images of the crystal structure of Na$_4$Fe$_3$(PO$_4$)$_2$(P$_2$O$_7$) material with three different types of Na$^+$ ions. **b** The migration energy barriers within the same Na$^+$ ion groups. **e** The migration energy barriers between different Na$^+$ ion groups (equivalent to three-dimensional diffusion pathways)

deionized water with magnetic stirring until a transparent solution was obtained (denoted as solution B). Then solution B was added dropwise to solution A with vigorous stirring. The mixed solution was then heated in a water bath at 90 °C until all the excess water was removed and the sol-gel precursor was obtained. The precursor was ground into a fine powder and then annealed at 500 °C for 24 h under high purity Ar gas atmosphere with an intermediate grinding. The final nanosized Na$_4$Fe$_3$(PO$_4$)$_2$(P$_2$O$_7$) plates were denoted as NFPP-E in this paper. For the microporous Na$_4$Fe$_3$(PO$_4$)$_2$(P$_2$O$_7$) particles, all the preparation procedures were the same, except that 0.6305 g citric acid monohydrate replaced the 0.8768 g ethylenediaminetetraacid acid in solution B. The final product was denoted as NFPP-C in this paper. Both the final NFPP-E and NFPP-C samples were transferred into an Ar-filled glove box after annealing until further use.

**Synthesis of polypyrrole (PPy)-coated Fe$_3$O$_4$ nanospheres**. The Fe$_3$O$_4$ nanospheres were prepared via a solvothermal method using trisodium citrate as stabilizer and FeCl$_3$ as raw material in ethylene glycol solution. Specifically, 4 mmol FeCl$_3$·6H$_2$O and 1.3 g trisodium citrate were dissolved in 30 ml ethylene glycol by stirring for 2 h; then, 0.28 sodium acetate was added into the solution. After 1 h of ultrasonication and 1 h of magnetic stirring, the yellow solution was transferred to and sealed in a Teflon-lined autoclave. The autoclave was maintained at 180 °C for 12 h and then cooled down to room temperature naturally. The Fe$_3$O$_4$ nanoparticles were obtained by centrifugation for several times with water and ethanol. Then, 30 mg of the as-obtained Fe$_3$O$_4$ nanospheres was added to 30 mL deionized water with paddle stirring in a three-neck round-bottom flask. 600 μL pyrrole was added to the solution, followed by 2 h of ultrasonication and 12 h of stirring. The final PPy-coated Fe$_3$O$_4$ nanospheres were obtained via freeze-drying under vacuum for 24 h.

**Materials characterization**. X-ray diffraction (XRD) was employed to characterize the crystalline structure of the obtained samples in the $2\theta$ range of 15°–45° (GBC MMA diffractometer, Cu Kα radiation, $\lambda = 1.5406$ Å, 1 ° min$^{-1}$, step size of 0.02 ° s$^{-1}$). Synchrotron powder diffraction data were collected at the Australian Synchrotron beamline at the wavelength ($\lambda$) of 0.7748 Å, calibrated with the

standard reference material (National Institute of Standards and Technology (NIST) LaB$_6$ 660b), and analysed via GSAS-II software[55]. Schematic representations of the synchrotron XRD data were obtained by VESTA software[56]. The morphologies of both samples were examined through a field emission scanning electron microscope (FESEM, JEOL JSM-7500FA). The detailed structural information on the crystals was determined by both high-resolution transmission electron microscopy (HRTEM, JEOL 2010, 200 kV) and scanning transmission electron microscopy (STEM, JEOL JEM-ARM200F, 200 kV) with EDS mapping and a selected area electron diffraction (SAED) module. The angular range of collected electrons for the HAADF images was around 70–250 mrad. The EDS mapping results were obtained via STEM using NSS software. Thermogravimetric (TG) analysis was performed on a Mettler Toledo TGA/DSC1 with a heating rate of 10 °C min$^{-1}$ in air. The surface information was collected via X-ray photoelectron spectroscopy (XPS) with Al Kα radiation. Raman spectra (JY HR800 spectrometer, 10 mW helium/neon laser at 632.8 nm) were collected to characterize the carbon of both samples. The vibration states of existing functional groups were examined via Fourier transform infrared spectroscopy (FT-IR, spectral resolution of 4 cm$^{-1}$ on a Thermo Nicolet Nexus 670 FT-IR spectrometer). The magnetic measurements were performed via a Quantum Design physical properties measurement system (PPMS) combined with a vibrating sample magnetometer (VSM) option at various temperature (5–300 K) from −20000 Oe to 20000 Oe. The Brunauer–Emmett–Teller (BET) testing was carried out using a Micromeritics Tristar II 3020 surface area analyzer. The valence states of Fe in the Na$_4$Fe$_3$(PO$_4$)$_2$(P$_2$O$_7$) were characterized via XANES analysis. The in-situ XRD was conducted at the P02.1 Powder Diffraction beamline and XANES spectra were recorded at the P64 XAS beamline of the DESY synchrotron. Atomic force microscopy (AFM) on the nanoplate particles was performed on an MPF-3D, Asylum Research, Santa Barbara, USA.

**Electrochemical testing**. The NFPP-E and NFPP-C electrodes were prepared by mixing 80 wt. % active material, 10 wt. % conductive carbon black, and 10 wt. % polyvinylidene fluoride (PVDF, binder) to form the electrode slurry with a proper amount of N-methyl-2-pyrrolidone (NMP) as solvent. The slurry was uniformly

coated on aluminium foil, followed by vacuum drying at 120 °C overnight. Both the $Fe_3O_4$ electrode and purchased hard carbon (KURARAY Co., Ltd., Japan (Type 2)) was prepared by mixing 80 wt. % active material, 10 wt. % conductive carbon black, and 10 wt. % carboxymethyl cellulose (CMC, binder) to form the electrode slurry with a proper amount of deionized water as solvent. The slurry was uniformly coated on copper foil, followed by vacuum drying at 80 °C overnight. The dried slurry on Al/Cu foil was pressed under 10 MPa and punched into small disc electrodes. The loading density was about 2.0 mg cm$^{-2}$ for the positive electrodes. The loading density of the negative electrodes was about 1.0–1.2 mg cm$^{-2}$. The weight ratio of the two electrodes was balanced with reference to the corresponding reversible capacities, and the current density is based on the anode mass. Na metal and $Fe_3O_4$/hard carbon electrodes were used as counter electrodes in the half cell and full-cell measurements, respectively. The electrolyte was 1 M $NaClO_4$ in ethylene carbonate (EC)–propylene carbonate (PC) solution (1:1 by volume) with 5 vol. % addition of fluoroethylene carbonate (FEC). Coin cells (CR2032) were assembled in a glove box filled with ultra-pure Ar, and the concentrations of $O_2$ and $H_2O$ were kept under 0.1 ppm. The electrochemical properties were examined by a NEWARE test system. The voltage window was set between 1.9 V–4.1 V, and 1 C = 120 mA g$^{-1}$. The high/low-temperature performance was investigated via employing a high/low-temperature test box (Si Yang Precision Equipment (Shanghai) Co. Ltd.). The low-temperature was set at −20 °C, and the high temperature was set at 50 °C. Electrochemical impedance spectroscopy (EIS) was employed in the range of 100 kHz to 10 mHz, with the amplitude of the AC voltage set at 10 mV (Bio-Logic VMP-3 electrochemical workstation). The ionic conductivity ($\sigma$) was obtained by Eq. (1)

$$\sigma = L/(R * S) \tag{1}$$

Where $L$ is the thickness of the electrodes, $R$ is the resistance tested by the same EIS mentioned above, and $S$ is the area of the electrode film. Cyclic voltammetry tests were carried out at various scanning speeds from 0.05 mV s$^{-1}$ to 0.4 mV s$^{-1}$ in the voltage window of 1.9 V–4.1 V. Galvanostatic intermittent titration technique (GITT) measurements were conducted on a Land battery testing system after 10 cycles to let the electrolyte/electrode reach its equilibrium state (current density: 6 mA g$^{-1}$, 0.05 C).

**Bond-valence calculations and DFT calculations**. The charge distribution and possible Na$^+$ ion motions were calculated using soft-BV parameters. First-principles calculations were performed using the Vienna *ab*-initio simulation package (VASP)[57], which is based on density functional theory and the plane-wave pseudopotential method[58], with the following settings. The generalized gradient approximation (GGA)[59] with the Perdew-Burke-Ernzerhof (PBE) exchange-correlation functional was used with the plane-wave cut-off energy set at 400 eV for all calculations. The criterion of convergence is that the residual forces should be less than 0.02 eV Å$^{-1}$ and the change in the total energy less than $10^{-5}$ eV. All calculations were performed using periodic boundary conditions, and the Brillouin-zone integrations were performed using a $3 \times 3 \times 3$ Monkhorst-Pack grid.

## Data availability

Data supporting the findings of this study are available from the authors on reasonable request. See author contributions for specific data sets.

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

## Acknowledgements

This work is supported by the Australian Research Council (ARC DP160102627) and the Australian Renewable Energy Agency (ARENA $S_4$) projects, the National Natural Science Foundation of China (Grant Nos. 11704114, 61427901, No. 21771164, U1804129), the Hunan Provincial Natural Science Foundation of China (Grant No. 2018JJ3110), the Scientific Research Fund of the Hunan Provincial Education Department of China (Grant No. 17C0462), and a China Postdoctoral Science Foundation Funded Project (Grant No. 2017M620872). The authors would like to thank Dr. Gilberto Casillas-Garcia for the STEM technique support and Dr. Tania Silver for critical reading of the manuscript. Parts of the experiments were carried out at the Powder Diffraction Beamline, Australian Synchrotron, and parts of the experiments were carried out at the P64 and P02.1 beamlines at the DESY Synchrotron, Hamburg, Germany.

## Author contributions

M. Chen prepared the manuscript. M. Chen and E. Wang synthesized the materials. Q. Gu conducted the powder diffraction. J. Xiao conducted the DFT studies, W. Hua and S. Indris carried out the in-situ XRD and in-situ XAS. D. Cortie and X. Wang performed the BVS calculations. Z. Hu conducted the AFM. W. Chen, S. Chou, and S. Dou supervised the project and co-wrote the manuscript. All authors discussed the results and contributed to writing the manuscript.

## Additional information

**Competing interests:** The authors declare no competing interests.

