## [Peer Review File · Nature Communications]

Reviewers' comments:

Reviewer #1 (Remarks to the Author):

I must admit the work is well carried out combining many sophisticated techniques. However, the manuscript lacks sufficient novelty for publication in Nature Comm.

1. In abstract and Introduction, you mention using earth-abundant element like Fe will reduce the cost. Think carefully....it is NOT true. There are two things: materials/precursor cost and processing cost. Fe is prone to oxidation. So we need to use Fe(II) based precursors which is not so cheap. More importantly, you need to anneal the Fe-based compounds in Ar-atmosphere to avoid oxidation. Making large-scale production using Ar-ambience is cumbersome and processing cost will be high (due to Ar). So, the net cost is not cheap. Fe-based compounds can be low-cost only if it is Fe(III) based composition (can be annealed in air) e.g. NaFeO_2 , $\text{Na}_x\text{Fe}_{1/2}\text{Mn}_{1/2}\text{O}_2$ (Komaba et al, Nature Mater). You can get economic system with Mn-analogue e.g. $\text{Na}_4\text{Mn}_3(\text{PO}_4)_2\text{P}_2\text{O}_7$. So, your claim of low-cost cathode (with Fe orthopyrophosphate) is WRONG.

2. This material is well known and well studied. So, materials point-of-view there is no novelty. The authors also agree with this fact.

3. Fig. 1a: XRD has large background. Should redo the scan with minimal background.

4. All characterisations (in Fig. 1) are very routine work. Its well performed but does not give any novel finding.

5. The structure refinement, magnetic data, XPS, Raman spectroscopy, BET are all very routine work with expected results. Nothing novel.

6. The TEM work is good, but again nothing never. All results are expected.

7. The electrochemical results (Fig 2) is again nothing extra-ordinary. Similar (better) results are reported earlier by Kisuk Kang group and others. e.g. ECS Transaction, 85(13), 207-214 (2018). ECS Transaction, 85(13), 227-234 (2018). You have constructed full cell, but the choice of Fe_2O_3 is not good (Fig 3). So, the full cell profile is very sloping and have low energy density.

8. The battery study at high and low temp is new. Good data presented.

9. Fig 4: Sodium storage mechanism is not novel. Its already published by Kang group (Chem Mater) and it is well known this system undergoes solid-solution (single phase) redox reaction. By doing synchrotron XRD and XAS, it does not show any new conclusion. It can be done with simple XRD machine. XAS shows FeII-FeIII valence change. What else do you expect? Its common sense. No novelty with this exotic technique.

10. Na-ion dynamics: This is also well studied and nothing new. Similar studies have been published by Kang group and Saiful Islam group (you have not cited their work). See 3 year old paper in 2015: Wood et al, J. Phys. Chem. C. 2015, 119, 15935-15941. So, this section is also Not novel.

Overall, the work is well carried out but lacks sufficient novelty. It can be submitted to some field journal like J Electrochemical Soc/ J Power Sources etc.

Reviewer #2 (Remarks to the Author):

The manuscript entitled "New NASICON-Type Air Stable and All-Climate Cathode for Sodium-Ion Batteries with Low-Cost and Long-Life High Power Density" reports a mixed-polyanionic based material for the cathode material for sodium-ion batteries, which is one of the important branches of the existing cathode materials. Various characterizations have been conducted to demonstrate its outstanding electrochemical properties and the reasons behind. The air stability and high/low temperature performance of this $\text{Na}_4\text{Fe}_3(\text{PO}_4)_2(\text{P}_2\text{O}_7)$ are collected, and adequate supporting data are displayed, which provides a wide-screen picture of this material and its potential to be large-scale manufactured at lower costs. Both the in-situ synchrotron-based XRD and the in-situ XANES spectra show the highly reversible nature regarding its excellent cycling stability. The authors also bring a new interesting concept of the NASICON-structure which is not discussed and recognised beyond the well-known ones such as $\text{Na}_3\text{V}_2(\text{PO}_4)_3$, and they performed the DFT study and BVS calculations to further explain their points. In my opinion, this manuscript will arise boarder attentions of researchers to this type materials and should be interesting to the readers. I recommend accepting it in Nature Communication after the following minor revisions.

--- According to the authors explanations, this $\text{Na}_4\text{Fe}_3(\text{PO}_4)_2(\text{P}_2\text{O}_7)$ possesses the 3D sodium diffusion pathways, thus it can be recognized as the NASICON-type material. It would be appropriate for the authors to discuss in details about both the same and/or different aspects to the well-known NASICON-type materials (such as $\text{Na}_3\text{V}_2(\text{PO}_4)_3$ and $\text{Na}_3\text{Ti}_2(\text{PO}_4)_3$).

--- There are several corrections required. (i) The XRD pattern in Figure 3a should be explained what wavelength was used and the XPS data in Figure 3b and c should be fitted to give more detailed information. (ii) The EIS data in Figure S9b and c are not fitted well and correctly. Please modify them.

--- Recent publications in Fe-based/polyanionic materials should be cited, compared and discussed appropriately as well.

--- Further explanation of Figure S4 (i.e. legends for red and blue colored lines and what they indicate) is needed. Also the FFT picture in Figure 4d should be indexed and labled. The elements distributions in FigureS12c and d are too small to be seen. Please make the fonts with proper size for readers.

Reviewer #3 (Remarks to the Author):

The manuscript from Chen et al. reported the development of a low cost scalable polyanionic cathode material for sodium-ion batteries. Different synthetic strategies were adapted to fabricate the cathode materials of tuneable morphology and particle size. Good rate capability and long-term cycling stability were demonstrated in both half cell and full cell designs. The air stability as well as the high/low temperature properties also were investigated. It is a big step for Na-ion battery development towards practical application. Furthermore, the authors conducted in-situ synchrotron XRD/XAS characterization and theoretical simulation to reveal the intrinsic nature of this material, which brings further in-depth understanding of the material. I believe the work is significant in the field and has potential broad impact. Hence, I recommend the paper to be published in Nature Communications. Below are some suggestions and minor questions for the authors to consider. Hopefully they will find them useful in further improving the paper quality.

1. Maybe I missed it but I did not find the rational of using Fe_3O_4 as the anode for the full cell demonstration. Hard carbon is usually the anode material for Na-ion battery full cells. I am curious why the authors did not choose that and what the performance will look like if hard carbon was used as anodes.
2. The full cell performance in figure 3 is really impressive. However, I just would like to remind the

author that it was cycled between 0 and 4V, which will make it not very practical because of the very low discharge cut off voltage.

3. The performance at low and high temperatures is impressive. For the test at -20C, I am wondering whether the cells were charged and discharge at the temperature. I would like to suggest the authors to add some detail information in the Method part.

4. With the excellent electrochemical performance, I am wondering whether the authors have tried high loading electrodes.

5. In Figure 2b, the CV curves are slightly different with 1st cycle and following cycles. Can the authors give more explanations about this phenomenon?

Responses to Reviewers:

We have carefully considered all the comments and questions raised by the reviewers. We took time to plan and carry out additional experiments, which helped to address the reviewers' comments and questions. Newly obtained data are included in the revised manuscript or supplementary information and the relevant discussions have been amended in the manuscript. We sincerely thank the reviewers for raising relevant questions and constructive comments which have, in our opinion, greatly helped us to improve the quality of the present work. The specific point to point responses are displayed below. Corresponding changes are marked in **turquoise** in the copy of the revised manuscript.

Reviewer #1 (Remarks to the Author):

I must admit the work is well carried out combining many sophisticated techniques. However, the manuscript lacks sufficient novelty for publication in Nature Comm.

1. In abstract and Introduction, you mention using earth-abundant element like Fe will reduce the cost. Think carefully....it is NOT true. There are two things: materials/precursor cost and processing cost. Fe is prone to oxidation. So we need to use Fe(II) based precursors which is not so cheap. More importantly, you need to anneal the Fe-based compounds in Ar-atmosphere to avoid oxidation. Making large-scale production using Ar-ambience is cumbersome and processing cost will be high (due to Ar). So, the net cost is not cheap. Fe-based compounds can be low-cost only if it is Fe(III) based composition (can be annealed in air) e.g. NaFeO_2 , $\text{Na}_x\text{Fe}_{1/2}\text{Mn}_{1/2}\text{O}_2$ (Komaba et al, Nature Mater). You can get economic system with Mn-analogue e.g. $\text{Na}_4\text{Mn}_3(\text{PO}_4)_2\text{P}_2\text{O}_7$. So, your claim of low-cost cathode (with Fe orthopyrophosphate) is WRONG.

Answer: Thanks for your profound comment. Indeed, iron is the most earth-abundant of the 3d elements, and it has been widely used since the beginning of human civilization. Since the sodium sources are almost unlimited, the utilization of Fe can further reduce the overall cost of electrode materials.³³ For large-scale production, the costs of the final cathode products comprise several part, such as the price of the raw material, energy consumption, etc. In the case of Fe-based layered oxide materials, almost all of them have to be sintered at high temperature (commonly above 850 °C).⁴⁵ Thus the heat consumption at high temperature (commonly generated by electricity) cannot be neglected. Meanwhile, in our manuscript, the $\text{Na}_4\text{Fe}_3(\text{PO}_4)_2(\text{P}_2\text{O}_7)$ material only underwent a 500 °C calcination process, which is considerably lower than that needed for the Fe-based layered oxides. On the other hand, those Fe-based layered oxide materials also face critical issues, such as their low capacity retention, continuous working voltage drop and capacity drop, air stability, initial cycle Coulombic efficiency, etc.³⁴ Polyanionic-based materials possess a robust crystallised framework, very stable voltage-capacity performance, and high initial cycle Coulombic efficiency, although their theoretical capacities are generally limited because of their large molecular weights.³⁴ Therefore, the polyanionic-based materials, especially Fe-based polyanionic materials, are excellent supplements to the various kinds of cathode materials in progress towards real applications of SIBs.

Another important issue that concerns the reviewer is the price of Fe(II) raw materials. Indeed, Fe is prone to oxidation, but we checked the prices on websites and found that the price of raw material containing only Fe(II) is just slightly higher than for those containing Fe(III). For instance, Fe (II) oxalate is common iron source for LiFePO_4 , and we checked the price – USD \$1760 per metric ton for Fe(II) oxalate and USD \$1500 per metric ton for Fe (III) oxalate (please refer to www.alibaba.com). Moreover, there are three aspects that the reviewer might have neglected.

(a) It is necessary to analyse the oxidizability under specific conditions. Fe^{2+} is easily oxidised to Fe^{3+} under aqueous conditions, but solid state Fe(II) raw materials are relatively stable and not easily oxidised when the Fe(II) raw materials are kept in a dry and inert atmosphere (such as N_2). N_2 is much cheaper compared to Ar (Please refer to www.boc.com.au). So, it will not cost much to carefully store these solid state Fe(II) raw material sources.

(b) In our manuscript, we used argon gas as the protective gas, since we consider that the high purity Ar gas can offer a completely inert protective atmosphere during sintering. In the real situation of industrial manufacturing, however, N_2 is normally adopted, it is generated from liquid nitrogen, and the purity of N_2 can reach 99.999 % or even higher. Also, since this $\text{Na}_4\text{Fe}_3(\text{PO}_4)_2(\text{P}_2\text{O}_7)$ material only needs a 500 °C sintering, there is no possibility for any reaction between the raw Fe sources and N_2 . In fact, the commercial LiFePO_4 material needs 600 °C or more to form the final phase, and N_2 is widely used in many world famous LIB factories. Nevertheless, we have redone the identical calcination procedure with the same precursor using high purity N_2 as the protective gas. The charge-discharge curve shows no difference from the one where Ar was used as the protective gas (Figure Ia and b). So, there is no need for reviewer to be concerned about the problems that Ar may bring.

(c) In iron-based polyanionic materials for both LIBs and SIBs, both iron (II) sources and iron (III) sources are applicable for synthesizing the final products, such as LiFePO_4 and $\text{Na}_{3.12}\text{Fe}_{2.44}(\text{P}_2\text{O}_7)_2$. The Fe (III) can be reduced to Fe (II) in the presence of carbon within an inert sintering atmosphere. So, we think that our $\text{Na}_4\text{Fe}_3(\text{PO}_4)_2(\text{P}_2\text{O}_7)$ material is also suitable for raw Fe (III) sources. We have redone the identical calcination procedure with the precursor using Iron (III) acetate as iron source (with the others identical as described in the Method part) and using high purity N_2 as the protective gas. Surprisingly, the charge-discharge curves showed no obvious difference from the ones using Fe (II) acetate (Figure Ia and c). Moreover, we assume that the other Fe-based polyanionic materials are amenable to sintering using raw Fe (III) sources, as long as the presence of carbon sources can be guaranteed during sintering (maybe at or above 500 °C).

Therefore, this $\text{Na}_4\text{Fe}_3(\text{PO}_4)_2(\text{P}_2\text{O}_7)$ material, or other Fe-based orthophosphate cathode, is really cheap and can be manufactured on a large scale at rather low-cost for the commercial SIB market.

Figure 1. First five cycles of as-prepared NFPP-E electrodes from different synthetic routes: (a) using Fe (II) acetate and Ar, (b) using Fe (II) acetate and N_2 , (c) using Fe (III) acetate and N_2 . Charge and discharge current: 0.1 C (12 mA g^{-1}).

2. This material is well known and well-studied. So, materials point-of-view there is no novelty. The authors also agree with this fact.

Answer: Thank you for your valuable comment. Kang's group firstly studied the $\text{Na}_4\text{Fe}_3(\text{PO}_4)_2\text{P}_2\text{O}_7$ material and obtained its structural details in 2012.^[1] Then, they characterized its physical and chemical properties with more techniques.^[2] Since then, there have been a few reports on this material from different aspects or perspectives, such as sodium ion diffusion and voltage trends,^[3] Mn-doping,^[4] potential application in aqueous systems,^[5] surface modifications,^[6,7] etc. Nevertheless, this material possesses a very stable charge/discharge curve, small volume shrinkage, and outstanding 3D sodium diffusion pathways, so it should receive more attention and more in-depth or comprehensive investigations apart from the reports mentioned above.

In our manuscript, our aim was to discover a polyanionic material that can be generally utilized over a wide range of temperatures. So, we initially focused on the electrodes that possess 3D sodium diffusion pathways. After screening the existing polyanionic material compounds, we

found that only two or three materials have 3D diffusion pathways. Therefore, it is meaningful to highlight their unique properties and provide a more complete investigation. Also we nanosized the material and gave the particles a uniform carbon coating, which is important for helping cathodes to demonstrate their best performance. More importantly, we tested its ionic conductivity, and we found that this material has a very similar value to that of the well-known sodium superionic conductor (NASICON)-type $\text{Na}_3\text{V}_2(\text{PO}_4)_3$ material. Considering its highly competitive ionic conductivity and 3D diffusion pathways, we think that it is appropriate to assign this material to the NASICON family, which will surely arouse extensive interest from researchers for more integrated investigations. Therefore, although this material has been previously reported, there is still considerable room for further explorations with wider applications, morphology control, and novel concepts.

3. Fig. 1a: XRD has large background. Should redo the scan with minimal background.

Answer: Thank you for your professional comment. We have redone the scans with a relatively flat background for both NFPP-E and NFPP-C materials. The revised Figure 1 and Figure S1 are shown below:

Figure 1a. Rietveld refinement of NFPP-E. A schematic representation of the refinement results is presented in the inset.

Figure S1. Rietveld refinement of NFPP-C sample. A schematic representation of the refinement results is presented in the inset.

4. All characterisations (in Fig. 1) are very routine work. It is well performed but does not give any novel finding.

Answer: Thank you for your valuable comment. The physical properties and nanoscale characterizations were carried out as shown in Figure 1. Synchrotron XRD is a powerful tool to perform Rietveld refinements with detailed atomic fractions, site positions, etc. Also, FT-IR, XPS, and Raman spectroscopy are the commonly used tools to characterize basic physical and chemical properties. These techniques are important to provide a whole picture of this material from different aspects. We found that the intensity ratio of the D band to the G band (I_D/I_G) of the NFPP-E sample (1.04) is larger than that of the NFPP-C sample (0.94), which indicates that the carbon conductivity of the NFPP-E particle surface would be higher than that for NFPP-C. Besides, by employing the sol-gel method with an appropriate complex agent, nanosized $\text{Na}_4\text{Fe}_3(\text{PO}_4)_2(\text{P}_2\text{O}_7)$ plates can be obtained, which can be verified by SEM and TEM tools. The AFM technique supports our findings in another dimension. The HAADF images firstly present the clear arrangements of atoms at the atomic level, which can help the readers to better understand this material *via* an authentic and visual approach. Therefore, although the characterisations in Figure 1 are commonly seen and routine work, they are providing

indispensable information on the as-obtained nanosized carbon coated $\text{Na}_4\text{Fe}_3(\text{PO}_4)_2(\text{P}_2\text{O}_7)$ material.

5. The structure refinement, magnetic data, XPS, Raman spectroscopy, BET are all very routine work with expected results. Nothing novel.

Answer: Thank you for your valuable concerns. Structure refinement is a basic and preliminary approach to provide detailed analysis of crystal structures. The magnetic data provides supplementary proof that can determine that whether there is any ferromagnetic or antiferromagnetic impurity in the main phase. Also, the effective magnetic moments can be calculated to detect the high/low spin state of iron in this material, which is important to further improve the voltage platform with appropriate element doping. The XPS, Raman spectroscopy, and BET techniques are offering a complete picture of the different properties of this as-prepared material. We think that these tests and data are basic and necessary. We have also refitted the XPS data with more proper Fe deconvolution curves of both samples, as shown below:

Figure S4. X-ray photoelectron spectroscopy (XPS) results on Fe for the (a) NFPP-E and (b) NFPP-C samples.

We cannot anticipate what we will get, however, until we run the tests. We need to use the routine work and techniques to obtain a deeper understanding step by step until we find

something novel and interesting. These thorough characterizations are required to obtain a deeper understanding of our as-prepared material.

6. The TEM work is good, but again nothing new. All results are expected.

Answer: Thank you for your valuable concerns. The electron transmission tests are based on scanning transmission electron microscopy, which is one of the most advanced techniques and involves state-of-the-art advances. The TEM work in Figure 1 and Figure S6 clearly show the lattice fringes and the atom array in the HAADF mode. It is very helpful for researchers who are not very familiar with this orthorhombic symmetry and also offers a visual method to more conveniently gain an understanding of the atomic arrangement. With the help of TEM, we can also better understand how the synthesis conditions influence the final morphology and the carbon coating situation. Also in Figure 4c and d, we look at the charged and discharged electrodes, and the lattice fringes can be clearly observed, which represent complete proof of the single phase transition during cycling. In addition, we employed EDS mapping to see whether there are any particular element-rich areas, and we found that both before and after cycling, charged and discharged samples all have a homogeneous element distribution within the nanosized particles. Thus, the obtained TEM related work is in a close accordance with the revealed electrochemical properties. It is important to employ TEM to characterise atomic level properties, and it is also a meaningful and innovative strategy to explore what we cannot see directly in the nano world.

7. The electrochemical results (Fig 2) is again nothing extra-ordinary. Similar (better) results are reported earlier by Kisuk Kang group and others.

e.g. ECS Transaction, 85(13), 207-214 (2018). ECS Transaction, 85(13), 227-234 (2018). You have constructed full cell, but the choice of Fe_2O_3 is not good (Fig 3). So, the full cell profile is very sloping and have low energy density.

Answer: Thanks for your insightful comment. Since this material possesses the 3D sodium diffusion pathways, outstanding rate performance can be achieved under various synthesizing conditions. Kang's group reported its preliminary electrochemical performance without further particle or morphology optimization.^{10,12} Since then, Islam's group explored the potential of other redox centres such as Ni and Co with calculated high voltage trends,¹¹ and Rojo's group investigated its performance in an aqueous system.¹⁵ Other researchers also did some relevant work on this material.^{16,17} Very recently, Barpanda's group explored its potential for thin-film SIBs and potassium intercalation possibilities.^{18,19} All the pioneering work is encouraging and shows the outstanding promise of this $\text{Na}_4\text{Fe}_3(\text{PO}_4)_2\text{P}_2\text{O}_7$ material for SIBs. We have cited their work in the revised manuscript. In our manuscript, we present the long-term cycling stability up to 4400 cycles, and we have compared all the Fe-based polyanionic cathode materials in **Figure S8**. We also emphasised the mid-working voltage retention, which is a vital parameter for energy retention that is often neglected by other researchers. We examined long-cycled electrodes, and good crystallinity was still well maintained. Based on these pioneering results, we developed a wider picture of its outstanding performance, as well as incorporating insights from GITT study and pseudocapacitance study. For cathode materials, these electrochemical results are meaningful and competitive with other researchers' results.

The reason for using Fe_3O_4 is that we initially were inspired by Prof. Khalil Amine's work published in *Nano Lett.*²⁰ They used Fe_3O_4 as the anode since hard carbon is severely limited by the applied current density. A high capacity over 250 mAh g^{-1} can be achieved at 0.05 C or 0.1 C, although, if the current density increases to 1 C or above, only less than 50 % capacity can be

obtained. The PPy-coated Fe_3O_4 nanospheres in our manuscript have shown relative good rate capability. Also the density of Fe_3O_4 is more than 2 times higher than for those hard carbons and amorphous carbons. As a result, the loading of Fe_3O_4 in the electrode is far higher than for hard carbon. This translates to a significant increase in the energy density of Fe_3O_4 on the cell level compared to hard carbon. So, we adapted the PPy-coated Fe_3O_4 in our manuscript even though its initial cycle Coulombic efficiency is relatively low. Nevertheless, we admit that hard carbon is more widely used to construct SIB full cells. So, we also fabricated full cells using purchased hard carbon (KURARAY Co., Ltd., Japan (Type 2)). SEM images and electrochemical properties of this hard carbon are shown below:

Figure S18 (a) and (b) SEM images of purchased hard carbon. (c) Initial three cycles of purchased hard carbon (100 mA g^{-1} , 0.4 C).

We then fabricated a $\text{Na}_4\text{Fe}_3(\text{PO}_4)_2(\text{P}_2\text{O}_7)$ /Hard carbon full cell with the loading mass ratio of 1.8 :1 to make the capacity balance according to the individual specific capacities of the electrodes. The anode electrodes are presodiated to reduce the dramatic initial irreversible capacity loss.

The electrochemical performance is displayed below:

Figure S18 (d) Initial 10 cycles of as-prepared $\text{Na}_4\text{Fe}_3(\text{PO}_4)_2(\text{P}_2\text{O}_7)/\text{Hard carbon}$ full cell within the voltage window from 0.5 V to 4.0 V at a current density of 100 mA g^{-1} . (e) Cyclability of as-prepared $\text{Na}_4\text{Fe}_3(\text{PO}_4)_2(\text{P}_2\text{O}_7)/\text{Hard carbon}$ full cell at a current density of 100 mA g^{-1} .

It can be seen that the charge/discharge curves of the $\text{Na}_4\text{Fe}_3(\text{PO}_4)_2(\text{P}_2\text{O}_7)/\text{Hard carbon}$ full cell are not as sloping as for those using PPy-coated Fe_3O_4 , so the energy density can be improved with an elevated mid-working voltage platform. Nevertheless, the capacity is continuously dropping within the initial 10 cycles. We cycled the fabricated full cell up to 120 cycles at 100 mA g^{-1} and found that the capacity retention is less than 50 %, and the charge capacity is always higher than discharge capacity for every cycle, which might be the main reason for the continuous capacity drop. The Coulombic efficiency of commercial hard carbon (97 % each cycle) is likely to be another reason for this. Nevertheless, we are still putting much effort into working towards better full cell performance using commercial hard carbon as anode with an optimised electrolyte system, more suitable loading ratio, etc. to be used in our future work.

We have added the corresponding figures and expressions to the revised manuscript. We hope that our reply can answer your questions and concerns.

8. The battery study at high and low temp is new. Good data presented.

Answer: Thank you for your positive encouragement. We focus on the real applications of SIBs, so we need to explore the all-weather performance of the cathode material. We will continuously test this sort of performance under various conditions.

9. Fig 4: Sodium storage mechanism is not novel. It's already published by Kang's group (Chem Mater) and it is well known this system undergoes solid-solution (single phase) redox reaction. By doing synchrotron XRD and XAS, it does not show any new conclusion. It can be done with a simple XRD machine. XAS shows FeII-FeIII valence change. What else do you expect? It's common sense. No novelty with this exotic technique.

Answer: Thank you for your critical comment. Indeed Kang's group has previously conducted *ex-situ* XRD testing during cycling, and they found that this system underwent a solid-solution redox reaction. It would be more appropriate and direct, however, to exhibit an *in-situ* test with consecutive variations. We examined the electrodes in both fully charged and fully discharged states with HRTEM, which shows detailed information on the crystal structure, as displayed in Figure 4c and d. We further confirmed the topotactic single phase transition process during the highly reversible cycling. In addition, we found that when the electrode is below the main discharge/charge platform, almost no two theta shifts can be observed, although after the main platform finishes, a relatively dramatic variation appears. The calculated lattice parameter shifts further reflect this phenomenon. Therefore, we think that this *in-situ* synchrotron-based XRD is of value to provide readers with more subtle relevant information. As for XAS, we also employed *in-situ* XAS, which provided the first reported such results, to the best of our knowledge. The remaining Na⁺ ions in the crystal structure (about 25 %) can be regarded as belonging to the binding pillars. We noticed that the spectra of both the initial state and discharge state were almost the same, but with an obvious discrepancy compared to the LiFePO₄ reference sample. This can be ascribed to the individual fingerprint information on P₂O₇ and PO₄ groups. Some of the oxidized Fe³⁺ remains at +3 and cannot be reversed back to Fe²⁺ upon discharging. Their

achievable reversibility is excellent, however. These phenomena could not be expected before these XAS experiments, so it was important and meaningful to conduct these experiments with the synchrotron radiation sources.

10. Na-ion dynamics: This is also well studied and nothing new. Similar studies have been published by Kang group and Saiful Islam group (you have not cited their work). See 3 year old paper in 2015: Wood et al, J. Phys. Chem. C. 2015, 119, 15935-15941. So, this section is also Not novel.

Answer: Thank you for your valuable comments. We have carefully looked at the published results from both Kang's group and Saiful Islam's group, and compared our findings to their results. We found that there are some novel discoveries and methods, as well as new concepts that will be interesting to the readers. We have cited their work in the revised manuscript. Firstly, we employed bond valence method (BVS) calculations for a preliminary examination of the ionic states and diffusion pathways, since it is a well-established tool against experiment. It was found that the isosurface near -3.5 eV (lowest energy regions) is all connected, and it is a very important representation for the 2D or 3D diffusion pathways (Figures S19-S20). These results provide strong support for conclusions of the following DFT study. Secondly, we found that all 16 sodium sites can be divided into three different types of sodium positions. Their specific E_f (binding energy) values have been summarised in Table S3. Unlike the report of Kang's group, we found that the lowest energy barrier is located on the *a*-axis. Besides, we provided detailed information on the energy barriers with the same and different sodium types, and the 3D diffusion pathways are clearly observed in Figure 5c and d. Thirdly, based on the 3D diffusion pathways and ionic conductivity test results, we consider that this type of material can be regarded as a new member of the existing family of NASICON-type materials. We believe that this new concept will throw light on the researchers' motivations to discover more 3D diffusion-enabled materials in the near future.

Overall, the work is well carried out but lacks sufficient novelty. It can be submitted to some field journal like J Electrochemical Soc/ J Power Sources etc.

Generally speaking, our manuscript reports tuneable nanosized $\text{Na}_4\text{Fe}_3(\text{PO}_4)_2\text{P}_2\text{O}_7$ plates *via* a facile one-step sol-gel method with high phase purity and uniform carbon coating. We also have conducted multiple physical and electrochemical tests, which has provided a comprehensive picture of this material with much potential to be manufactured on a large scale. In addition, the low/high temperature testing and air stability investigation have brought more exciting properties to light, and both the BVS calculations and the DFT study clearly show the 3D sodium diffusion pathways, indicating its high rate capability resulting from various synthetic conditions. It is important to expand the family of the well-known NASICON-type materials because of its competitive ionic conductivity and 3D low energy barriers. We believe that our research will throw light upon the polyanionic-based materials and attract wide attention from more researchers. All the new aspects and characterizations mentioned above point to the sufficient novelty of our study. Hence, the authors think that this study should be published in *Nature Communications*. We really appreciate the overall reviewing work that has taken up your precious time, and we hope that our revised manuscript will answer your questions and meet your requirements. Thank you again for your reviewing work and patience.

- 1 Choi, J. W. & Aurbach, D. Promise and reality of post-lithium-ion batteries with high energy densities. *Nat. Rev. Mater.* **1**, 16013 (2016).
- 2 Fang, Y. *et al.* Recent Progress in Iron-Based Electrode Materials for Grid-Scale Sodium-Ion Batteries. *Small* **14**, 1703116 (2018).
- 3 Barpanda, P., Oyama, G., Nishimura, S., Chung, S. C. & Yamada, A. A 3.8-V earth-abundant sodium battery electrode. *Nature Commun.* **5**, 4358, (2014).
- 4 Yabuuchi, N. *et al.* P2-type $\text{Na}_{(x)}[\text{Fe}_{(1/2)}\text{Mn}_{(1/2)}]\text{O}_2$ made from earth-abundant elements for rechargeable Na batteries. *Nat. Mater.* **11**, 512-517, (2012).

- 5 Tapia-Ruiz, N. *et al.* High voltage structural evolution and enhanced Na-ion diffusion in P2- $\text{Na}_{2/3}\text{Ni}_{1/3-x}\text{Mg}_x\text{Mn}_{2/3}\text{O}_2$ ($0 \leq x \leq 0.2$) cathodes from diffraction, electrochemical, and ab initio studies. *Energy Environ. Sci.* **11**, 1470-1479 (2018).
- 6 Yuan, Y., Amine, K., Lu, J. & Shahbazian-Yassar, R. Understanding materials challenges for rechargeable ion batteries with in situ transmission electron microscopy. *Nat. Commun.* **8**, 15806 (2017).
- 7 Oh, S.-M. *et al.* High Capacity O3-Type $\text{Na}[\text{Li}_{0.05}(\text{Ni}_{0.25}\text{Fe}_{0.25}\text{Mn}_{0.5})_{0.95}]\text{O}_2$ Cathode for Sodium Ion Batteries. *Chem. Mater.* **26**, 6165-6171 (2014).
- 8 Kubota, K. *et al.* Understanding the Structural Evolution and Redox Mechanism of a NaFeO_2 - NaCoO_2 Solid Solution for Sodium-Ion Batteries. *Adv. Funct. Mater.* **26**, 6047-6059 (2016).
- 9 Lee, E. *et al.* New Insights into the Performance Degradation of Fe-Based Layered Oxides in Sodium-Ion Batteries: Instability of $\text{Fe}^{3+}/\text{Fe}^{4+}$ Redox in $\alpha\text{-NaFeO}_2$. *Chem. Mater.* **27**, 6755-6764 (2015).
- 10 Barpanda, P., Lander, L., Nishimura, S.-i. & Yamada, A. Polyanionic Insertion Materials for Sodium-Ion Batteries. *Adv. Energy Mater.* **8**, 1703055 (2018).
- 11 Kim, H. *et al.* New iron-based mixed-polyanion cathodes for lithium and sodium rechargeable batteries: Combined first principles calculations and experimental study. *J Am. Chem. Soc.* **134**, 10369-10372 (2012).
- 12 Kim, H. *et al.* Understanding the Electrochemical Mechanism of the New Iron-Based Mixed-Phosphate $\text{Na}_4\text{Fe}_3(\text{PO}_4)_2(\text{P}_2\text{O}_7)$ in a Na Rechargeable Battery. *Chem. Mater.* **25**, 3614-3622, (2013).
- 13 Wood, S. M., Eames, C., Kendrick, E. & Islam, M. S. Sodium Ion Diffusion and Voltage Trends in Phosphates $\text{Na}_4\text{M}_3(\text{PO}_4)_2\text{P}_2\text{O}_7$ ($\text{M} = \text{Fe}, \text{Mn}, \text{Co}, \text{Ni}$) for Possible High-Rate Cathodes. *J Phys. Chem. C* **119**, 15935-15941 (2015).
- 14 Kim, H. *et al.* Highly Stable Iron- and Manganese-Based Cathodes for Long-Lasting Sodium Rechargeable Batteries. *Chem. Mater.* **28**, 7241-7249 (2016).
- 15 Fernández-Ropero, A. J., Zarrabeitia, M., Reynaud, M., Rojo, T. & Casas-Cabanas, M. Toward Safe and Sustainable Batteries: $\text{Na}_4\text{Fe}_3(\text{PO}_4)_2\text{P}_2\text{O}_7$ as a Low-Cost Cathode for Rechargeable Aqueous Na-Ion Batteries. *J Phys. Chem. C* **122**, 133-142 (2017).
- 16 Wu, X., Zhong, G. & Yang, Y. Sol-gel synthesis of $\text{Na}_4\text{Fe}_3(\text{PO}_4)_2(\text{P}_2\text{O}_7)/\text{C}$ nanocomposite for sodium ion batteries and new insights into microstructural evolution during sodium extraction. *J Power Sources* **327**, 666-674 (2016).
- 17 Kosova, N. V. & Belotserkovsky, V. A. Sodium and mixed sodium/lithium iron orthopyrophosphates: Synthesis, structure and electrochemical properties. *Electrochim. Acta* **278**, 182-195 (2018).

- 18 S. Baskar, R. A., C. Murugesan, S. B. Krupanidhi, and P. Barpanda. Exploration of Iron-Based Mixed Polyanion Cathode Material for Thin-film Sodium-ion Batteries. *ECS Trans.* **85**, 227-234 (2018).
- 19 K. Sada, C. M., S. Baskar, and P. Barpanda. Potassium Intercalation into Sodium Metal Oxide and Polyanionic Hosts: Few Case Studies. *ECS Trans.* **85**, 207-214 (2018).
- 20 Oh, S. M. *et al.* Advanced Na[Ni_{0.25}Fe_{0.5}Mn_{0.25}]O₂/C-Fe₃O₄ sodium-ion batteries using EMS electrolyte for energy storage. *Nano Lett.* **14**, 1620-1626 (2014).

Reviewer #2 (Remarks to the Author):

The manuscript entitled “New NASICON-Type Air Stable and All-Climate Cathode for Sodium-Ion Batteries with Low-Cost and Long-Life High Power Density” reports a mixed-polyanionic based material for the cathode material for sodium-ion batteries, which is one of the important branches of the existing cathode materials. Various characterizations have been conducted to demonstrate its outstanding electrochemical properties and the reasons behind. The air stability and high/low temperature performance of this $\text{Na}_4\text{Fe}_3(\text{PO}_4)_2(\text{P}_2\text{O}_7)$ are collected, and adequate supporting data are displayed, which provides a wide-screen picture of this material and its potential to be large-scale manufactured at lower costs. Both the in-situ synchrotron-based XRD and the in-situ XANES spectra show the highly reversible nature regarding its excellent cycling stability. The authors also bring a new interesting concept of the NASICON-structure which is not discussed and recognised beyond the well-known ones such as $\text{Na}_3\text{V}_2(\text{PO}_4)_3$, and they performed the DFT study and BVS calculations to further explain their points. In my opinion, this manuscript will arise boarder attentions of researchers to this type materials and should be interesting to the readers. I recommend accepting it in Nature Communication after the following minor revisions.

1. According to the authors explanations, this $\text{Na}_4\text{Fe}_3(\text{PO}_4)_2(\text{P}_2\text{O}_7)$ possesses the 3D sodium diffusion pathways, thus it can be recognized as the NASICON-type material. It would be appropriate for the authors to discuss in details about both the same and/or different aspects to the well-known NASICON-type materials (such as $\text{Na}_3\text{V}_2(\text{PO}_4)_3$ and $\text{Na}_3\text{Ti}_2(\text{PO}_4)_3$).

Answer: Thank you for your insightful comment. NASICON is short for sodium super ionic conductor, which was first defined and studied by Hong, Goodenough, and Kafalas.^{1,2} They both synthesized $\text{Na}_{1+x}\text{Zr}_2\text{SixP}_{3-x}\text{O}_{12}$ material and analysed it in detail, and they both found that it is

very suitable for solid electrolyte with unexpectedly high ionic conductivity. It was also very rapidly recognized that other transition metal elements could be placed as redox centres in the structure, resulting in the expanded family of the sodium super ionic conductors. The generally well-known formula of NASICON materials can be written as $\text{Na}_x\text{MM}'(\text{XO}_4)_3$ ($\text{M} = \text{V}, \text{Ti}, \text{Fe}$, etc.; $\text{X} = \text{P}$ or S , $x = 0$ to 4). The NASICON framework is constructed from corner-shared MO_6 ($\text{M}'\text{O}_6$) octahedra and XO_4 tetrahedra. Some cathodes such as $\text{Na}_3\text{V}_2(\text{PO}_4)_3$,³ $\text{Na}_3\text{FeV}(\text{PO}_4)_3$,⁴ and $\text{Na}_4\text{MnV}(\text{PO}_4)_3$,⁴ have been extensively explored within the rigorous definition of NASICON structure mentioned above; however, some other derived formulas, such as $\text{Na}_3\text{V}_2(\text{PO}_4)_2\text{F}_3$,⁵ $\text{Na}_3(\text{VO}_{0.5})_2(\text{PO}_4)_2\text{F}_2$,⁶ $\text{Na}_3\text{V}_2\text{O}_{2x}(\text{PO}_4)_2\text{F}_{3-2x}$ ($x = 0$ to 1),⁷ smoothly spread the definition of NASICON structure. From our point of view, these types of materials possess very similar physical and chemical properties. The PO_4 or SO_4 tetrahedra keep the charge balance and interact with bipolar $\text{MO}_6\text{F}_{6-x}$ octahedra, while the $\text{MO}_6\text{F}_{6-x}$ octahedra can offer the main electrostatic repulsion that accounts for the various Na^+ de-/intercalation voltage platforms. The critical factor for a NASICON-type structure is the arrangement of sodium sites and whether the 3D diffusion pathways of Na^+ can be achieved with relatively low energy barriers. As for $\text{Na}_4\text{Fe}_3(\text{PO}_4)_2(\text{P}_2\text{O}_7)$ material, the P_2O_7 ditetrahedra are constituted of two corner-linked PO_4 tetrahedra, and the PO_4 tetrahedra share edges with the FeO_6 octahedra. This structure is very similar to those pure NASICON-type structures, and all the sodium ions are located in all the connected channels that enable 3D diffusion, as illustrated in our manuscript. Therefore, we consider that it is appropriate and necessary to expand the family of the NASICON materials. Moreover, we are aiming to add more families of the polyanionic-based materials to the expanded family of existing NASICON structures, so long as they can meet the requirements mentioned above. We also added the corresponding discussion to the revised manuscript.

2. There are several corrections required. (i) The XRD pattern in Figure 3a should be explained what wavelength was used and the XPS data in Figure 3b and c should be fitted to give more detailed information. (ii) The EIS data in Figure S9b and c are not fitted well and correctly. Please modify them.

Answer: Thank you for your valuable suggestions. We are sorry for the inaccuracies in the manuscripts. We have revised the manuscript according to your comments.

Figure 3a has been modified as shown below:

Figure 3 (a) XRD comparison of NFPP-E powder in the fresh state and after exposure to air for three months (Cu K α radiation, $\lambda = 1.5406 \text{ \AA}$).

Figure 3b has been modified as shown below:

Figure 3 (b) Fe 2p, C 1s, and P 2p XPS fitted spectra of NFPP-E powder in the fresh state and after exposure to air for three months.

Figure S9b and c have been modified as shown below:

Figure S9 EIS spectra of NFPP-E electrode (b) before cycling and (c) after the first cycle. The insets are the equivalent circuits used for interpreting the data. CPE: constant phase element, W: Warburg impedance.

3. Recent publications in Fe-based/polyanionic materials should be cited, compared and discussed appropriately as well.

Answer: Thank you for this valuable comment. We have added the recently published papers relevant to the $\text{Na}_4\text{Fe}_3(\text{PO}_4)_2(\text{P}_2\text{O}_7)$ material (Refw. 50-54). We are glad to see that this material is being paid more and more attention by researchers worldwide.

4. Further explanation of Figure S4 (i.e. legends for red and blue colored lines and what they indicate) is needed. Also the FFT picture in Figure 3d should be indexed and labeled. The elements distributions in FigureS12c and d are too small to be seen. Please make the fonts with proper size for readers.

Answer: Thank you for your professional suggestions. We have revised the manuscript according to your comments.

Figure S4 has been modified as shown below:

Figure S4 X-ray photoelectron spectroscopy (XPS) results on Fe for the (a) NFPP-E and (b) NFPP-C samples.

Figure 3d has been modified as shown below:

Figure 3(d) HRTEM image of the powder after exposure to air for three months. The inset is the fast Fourier transform (FFT) pattern of the marked area.

Figure S12c and d have been modified as shown below:

Figure S12 EDS spectra of (c) NFPP-E and (d) NFPP-C after 4400 cycles.

- 1 Hong, H. Y. P. Crystal structures and crystal chemistry in the system $\text{Na}_{1+x}\text{Zr}_2\text{Si}_x\text{P}_{3-x}\text{O}_{12}$. *Mater. Res. Bull.* **11**, 173-182 (1976).
- 2 J. B. Goodenough, H. Y.-P. H., J. A. Kafalas. Fast Na^+ -ion Transport in skeleton structures. *Mater. Res. Bull.* **11**, 203-220 (1976).
- 3 Zhu, C., Song, K., van Aken, P. A., Maier, J. & Yu, Y. Carbon-coated $\text{Na}_3\text{V}_2(\text{PO}_4)_3$ embedded in porous carbon matrix: An ultrafast Na-storage cathode with the potential of outperforming Li cathodes. *Nano Lett.* **14**, 2175-2180 (2014).
- 4 Zhou, W. *et al.* $\text{Na}_x\text{MV}(\text{PO}_4)_3$ (M = Mn, Fe, Ni) Structure and Properties for Sodium Extraction. *Nano Lett.* **16**, 7836-7841 (2016).
- 5 Zhu, C. *et al.* A High Power–High Energy $\text{Na}_3\text{V}_2(\text{PO}_4)_2\text{F}_3$ Sodium Cathode: Investigation of Transport Parameters, Rational Design and Realization. *Chem. Mater.* **29**, 5207-5215 (2017).
- 6 Xiang, X., Lu, Q., Han, M. & Chen, J. Superior high-rate capability of $\text{Na}_3(\text{VO}_{0.5})_2(\text{PO}_4)_2\text{F}_2$ nanoparticles embedded in porous graphene through the pseudocapacitive effect. *Chem. Commun.* **52**, 3653-3656 (2016).
- 7 P. Ramesh Kumar, Y. H. J., Brindha Moorthy, Do Kyung Kim. Effect of Electrolyte Additives on $\text{NaTi}_2(\text{PO}_4)_3\text{-C}/\text{Na}_3\text{V}_2\text{O}_{2x}(\text{PO}_4)_2\text{F}_{3-2x}\text{-MWCNT}$ Aqueous Rechargeable Sodium Ion Battery Performance. *J. Electrochem. Soc.* **163**, A1484-A1492 (2016).

Reviewer #3 (Remarks to the Author):

The manuscript from Chen *et al.* reported the development of a low cost scalable polyanionic cathode material for sodium-ion batteries. Different synthetic strategies were adapted to fabricate the cathode materials of tuneable morphology and particle size. Good rate capability and long-term cycling stability were demonstrated in both half cell and full cell designs. The air stability as well as the high/low temperature properties also were investigated. It is a big step for Na-ion battery development towards practical application. Furthermore, the authors conducted *in-situ* synchrotron XRD/XAS characterization and theoretical simulation to reveal the intrinsic nature of this material, which brings further in-depth understanding of the material. I believe the work is significant in the field and has potential broad impact. Hence, I recommend the paper to be published in Nature Communications.

Below are some suggestions and minor questions for the authors to consider. Hopefully they will find them useful in further improving the paper quality.

1. Maybe I missed it but I did not find the rational of using Fe_3O_4 as the anode for the full cell demonstration. Hard carbon is usually the anode material for Na-ion battery full cells. I am curious why the authors did not choose that and what the performance will look like if hard carbon was used as anodes.

Answer: Thanks for your professional question. Indeed, hard carbon is usually the anode material for fabricating full SIBs. Initially we were inspired by Prof. Khalil Amine's work published in Nano Lett.,¹ where they used Fe_3O_4 as the anode, since hard carbon is severely limited by the applied current density. A high capacity over 250 mAh g⁻¹ can be achieved at 0.05 C or 0.1 C, but if the current density increases to 1 C or above, only less than 50 % capacity can be obtained. The PPy-coated Fe_3O_4 nanospheres in our manuscript have shown relatively good rate capability. Also

the density of Fe_3O_4 is more than 2 times higher than for hard carbons and amorphous carbons. As a result, the loading of Fe_3O_4 in the electrode is much higher than for hard carbon. This translates to a significant increase in the energy density of Fe_3O_4 on the cell level compared to hard carbon. So, we adopted the PPy-coated Fe_3O_4 is our manuscript, although its initial cycle Coulombic efficiency is relatively low. Nevertheless, we agree that hard carbon is the more common choice for SIBs. We fabricated the cells using hard carbon as anode. The anodes were purchased from KURARAY Co., Ltd., Japan (Type 2). The SEM images and electrochemical properties of this hard carbon are shown below:

Figure S18 (a) and (b) SEM images of purchased hard carbon. (c) Initial three cycles of purchased hard carbon (100 mA g^{-1} , 0.4 C).

We then fabricated the $\text{Na}_4\text{Fe}_3(\text{PO}_4)_2(\text{P}_2\text{O}_7)$ //Hard carbon full cell with the loading mass ratio of 1.8 :1 to make the capacity balance according to the individual specific capacities of the electrodes. The anode electrodes were presodiated to reduce the dramatic initial irreversible capacity loss.

The electrochemical performance is displayed below:

Figure S18 (d) Initial 10 cycles of as-prepared $\text{Na}_4\text{Fe}_3(\text{PO}_4)_2(\text{P}_2\text{O}_7)/\text{Hard carbon}$ full cell within the voltage window from 0.5 V to 4.0 V at a current density of 100 mA g^{-1} . (e) Cyclability of as-prepared $\text{Na}_4\text{Fe}_3(\text{PO}_4)_2(\text{P}_2\text{O}_7)/\text{Hard carbon}$ full cell at a current density of 100 mA g^{-1} .

It can be seen that the charge/discharge curves of the $\text{Na}_4\text{Fe}_3(\text{PO}_4)_2(\text{P}_2\text{O}_7)/\text{Hard carbon}$ full cell are not as sloping as those using PPy-coated Fe_3O_4 , so the energy density can be improved with an elevated mid-working voltage platform, although the capacity is continuously dropping within the initial 10 cycles. We also cycled the fabricated full cell for 120 cycles at 100 mA g^{-1} , finding that the capacity retention is less than 50 %, and the charge capacity is always higher than discharge capacity for every cycle, which might be the main reason for the continuous capacity drop. The Coulombic efficiency of commercial hard carbon (97 % each cycle) is likely to be another reason for this. Nevertheless, we are still putting much effort into achieving better full cell performance using commercial hard carbon as anode with an optimised electrolyte system, more suitable loading ratio, etc. in our future work.

We have added the corresponding figures and expressions in the revised manuscript. We hope that our reply can answer your questions and concerns.

2. The full cell performance in figure 3 is really impressive. However, I just would like to remind

the author that it was cycled between 0 and 4V, which will make it not very practical because of the very low discharge cut off voltage.

Answer: Thanks for your professional comment. We are sorry that we did not notice that if we selected the very low discharge cut off voltage, the practical value would be reduced due to the required voltage range. We fabricated the full cells using hard carbon as anode with an appropriate voltage window (4.0 V- 0.5 V). We will keep this issue in mind in our further work, including on other types of electrodes when making full cells for SIBs.

3. The performance at low and high temperatures is impressive. For the test at $-20\text{ }^{\circ}\text{C}$, I am wondering whether the cells were charged and discharge at the temperature. I would like to suggest the authors to add some detail information in the Method part.

Answer: Thank you for your valuable comments. We apologize for the missing explanations of high/low temperature tests in the Methods part. We have modified the manuscript according to your comments. In order to remove your doubt, we have retested the cells at $-20\text{ }^{\circ}\text{C}$ using the high/low temperature test box. We have retested four cells, and the data are summarized below:

Figure II. (a) Low temperature performance of four individual cells containing NFPP-E electrodes at $-20\text{ }^{\circ}\text{C}$ and their corresponding charge-discharge curves at 1 C.

It can be seen that there are some deviations for the four individual cells, although similar capacity values to the data presented at Figure 3e can be achieved at various current densities (Figure IIa). Also we have plotted the corresponding charge-discharge curves for the NFPP-E electrode at 1 C, and around 60 mAh g⁻¹ can be obtained at -20 °C (Figure IIb). We have revised Figure S16 with the newly measured average curves:

Figure S16 Comparison of the charge/discharge curves of NFPP-E electrodes at room temperature, 50°C, and -20 °C, respectively.

We also added the recently published papers to the revised manuscript, although they have obtained similar results.² We hope that our added work and explanations can remove your doubts.

4. With the excellent electrochemical performance, I am wondering whether the authors have tried high loading electrodes.

Answer: Thank you for this practical concern. The loading density in the manuscript is about 2.0 mg cm⁻², which is in the middle among the reported electrochemical results for various positive electrodes. To address more practical concerns, we increased the loading mass to ~3.5 mg cm⁻², and the corresponding electrochemical performance is shown below:

Figure III. (a) C-Rate comparison of different loading masses. (b) Charge-discharge curves for higher loading mass of 3.5 mg cm^{-2} .

It can be seen that with the increased loading mass, the NFPP-E electrodes ($\sim 3.5 \text{ mg cm}^{-2}$) showed slightly decreased C-rate capacities compared to the lower one (2.0 mg cm^{-2}) (Fig. IIIa). Almost no obvious discrepancy can be seen at small current densities (0.1 C and 0.2 C), meaning that the full capacity can be achieved at tiny (negligible) electrochemical polarizations. With high current densities, however, large polarizations can be seen in Fig. IIIb, since the thicker electrode will result in longer distances for sodium ion transport. The noticeable capacity drop is reasonable and predictable, and around 77 mAh g^{-1} still can be obtained with this high loading mass. We will remember to increase the loading mass on prepared electrodes in our further or other works.

5. In Figure 2b, the CV curves are slightly different with 1st cycle and following cycles. Can the authors give more explanations about this phenomenon?

Answer: Thank you for your professional question. In Figure 2b, we also noticed that there is a small difference between the first cycle and the following cycles. Generally speaking, this phenomenon can be ascribed to several main reasons: (i). Phase transition during the first charge/discharge process. Some of other polyanionic-based material will undergo this phase transition during the first charge process, such as alluaudite-type $\text{Na}_2\text{Fe}_2(\text{SO}_4)_3$ material^{3,4} and

Na₂FeP₂O₇ material.⁵ Obvious peak shape changes can be seen directly. These phase transitions are normally driven by the strong electrostatic repulsion created *via* edge-sharing Fe₂O₁₀ dimers. (ii). The formation of the solid-electrolyte interphase (SEI) layer during the first cycle. The decomposition of electrolyte and side reactions take place in the first cycle, forming a consolidated SEI layer to buffer the subsequent cycles. For this reason, normally there is no peak shape change, while just detectable peak position shifts appear. (iii). Different scan rates. A larger scan rate will result in a stronger current response (more non-faradaic contribution to the total current), whereas some tiny peaks will not be seen due to the generation of larger polarization or stronger double-layer capacitance. In our material in the manuscript, it can be seen that there are no obvious peak shape changes (both in Figure 2b and Figure S9a), and only small peak position shifts can be detected. We also used the same scan rate (0.05 mV s⁻¹), so the main reason for the peak discrepancy is the formation of the SEI layer during first charge process. In order to further confirm our proposed reasons, we present the charge-discharge curves of the first two cycles and the corresponding dQ/dV plots (Figure IVa and b).

Figure IV (a) Charge and discharge curves of the first cycle and second cycle of NFPP-E electrode, and (b) corresponding dQ/dV plots.

Clearly the peaks have the same shape with small position shifts. So, we believe that there is no detectable phase transition during the first/second cycle apart from the SEI layer formation, which is the dominant reason for the slight difference between the 1st cycle and following cycles.

We have added some brief explanations to the revised manuscript. We sincerely hope that our explanations can answer your questions.

- 1 Oh, S. M. *et al.* Advanced Na[Ni_{0.25}Fe_{0.5}Mn_{0.25}]O₂/C-Fe₃O₄ sodium-ion batteries using EMS electrolyte for energy storage. *Nano Lett.* **14**, 1620-1626 (2014).
- 2 Ma, X. D., Wu, X. H. & Shen, P. K. Rational Design of Na₄Fe₃(PO₄)₂(P₂O₇) Nanoparticles Embedded in Graphene: Toward Fast Sodium Storage Through the Pseudocapacitive Effect. *ACS Appl. Energy Mater.* **1**, 6268-6278 (2018).
- 3 Oyama, G. *et al.* Sodium Intercalation Mechanism of 3.8 V Class Alluaudite Sodium Iron Sulfate. *Chem. Mater.* **28**, 5321-5328 (2016).
- 4 Chen, M. Z. *et al.* A Novel Graphene Oxide Wrapped Na₂Fe₂(SO₄)₃/C Cathode Composite for Long Life and High Energy Density Sodium-Ion Batteries. *Adv. Energy Mater.* **8**, 1800944, (2018).
- 5 Kim, H. *et al.* Na₂FeP₂O₇ as a Promising Iron-Based Pyrophosphate Cathode for Sodium Rechargeable Batteries: A Combined Experimental and Theoretical Study. *Adv. Funct. Mater.* **23**, 1147-1155 (2013).

REVIEWERS' COMMENTS:

Reviewer #1 (Remarks to the Author):

Good attempt on rebuttal.

Reviewer #2 (Remarks to the Author):

The revised manuscript has been improved and polished well according to the comments from all three reviewers. I recommend it acceptance as it is.

Reviewer #3 (Remarks to the Author):

I can see the authors have thoroughly revised the paper and addressed the questions, not only the ones from mine but also other reviewers'. The paper is important in the development of Na-ion batteries considering high energy density with novel anode and performance under different temperatures and other environmental conditions. I will recommend it to be published.

Point by Point Responses:

Reviewer #1 (Remarks to the Author):

Good attempt on rebuttal.

Response: We highly appreciate the reviewer's positive comments.

Reviewer #2 (Remarks to the Author):

The revised manuscript has been improved and polished well according to the comments from all three reviewers. I recommend it acceptance as it is.

Response: We highly appreciate the reviewer's positive comments.

Reviewer #3 (Remarks to the Author):

I can see the authors have thoroughly revised the paper and addressed the questions, not only the ones from mine but also other reviewers'. The paper is important in the development of Na-ion batteries considering high energy density with novel anode and performance under different temperatures and other environmental conditions. I will recommend it to be published.

Response: We highly appreciate the reviewer's positive comments.